

# The interaction of warm conveyor belt outflows with the upper-level waveguide: a four-type climatological classification

Selvakumar Vishnupriya[1], Michael Sprenger[1], Hanna Joos[1], and Heini Wernli[1]

[1]Institute for Atmospheric and Climate Science, ETH Zürich

**Correspondence:** Selvakumar Vishnupriya (vishnupriya.selvakumar@env.ethz.ch)

**Abstract.** Warm conveyor belts (WCBs) are coherent airstreams in extratropical cyclones, characterized by rapid ascent, intense latent heating, and cross-isentropic flow, reaching upper-tropospheric levels in their outflow. The divergent outflow of the WCB with low potential vorticity (PV) influences the upper-level PV distribution, thereby modifying the Rossby waveguide and amplifying the non-linear flow evolution. For instance, the interactions of WCB outflows with the waveguide can initiate the formation of blocks and of Rossby wave breaking, potentially leading to high-impact weather events in the regions of the interaction and downstream.

This study introduces a diagnostic approach to classify the WCB-waveguide interactions into four distinct types based on the intensity of the WCB-related waveguide disturbance: (i) weak/no interaction, (ii) ridge, (iii) block, and (iv) tropospheric cutoff interactions. Using ERA5 reanalysis data, we present the first systematic climatology (1980-2022) of the different interaction types, quantifying their frequency and the environmental conditions. The Lagrangian method is based on five-day backward trajectories from the upper tropospheric waveguide region, which fulfill typical WCB criteria. They are classified into different interaction types based on the presence of ridges, blocks, and cutoffs at their starting points. The method is applied globally and in all seasons, but this paper focuses mainly on the Northern Hemisphere winter (DJF).

The WCB identification and interaction classification method is illustrated first for previously documented cases of WCB outflows that influenced the upper-level dynamics. The climatological analysis in DJF shows that WCB outflows most frequently lead to ridge interaction (58.7%), followed by no interaction (27.7%), and rarely proceed to block and cutoff interactions (9.7% and 3.9%, respectively), with each interaction type occurring in preferred regions. The climatology highlights that the latitude of the WCB outflow and end-of-ascent clearly differ between the interaction types, whereas the latitudinal distribution of the WCB inflow and the start-of-ascent is fairly similar across the four types. As the intensity of the interaction increases from type (i) to (iv), the associated WCB outflows occur further poleward and westward, have a stronger negative PV anomaly, and reach lower pressure levels. The preceding ambient large-scale flow conditions also significantly differ between the interaction types, indicating the large influence of the preexisting synoptic flow situation on how WCBs interact with the upper-level waveguide. Weak/no interactions occur in situations with weak synoptic activity and an undisturbed zonally oriented waveguide, while the intense interactions are typically preceded by upper-level ridges and strong synoptic activity.



## 1 Introduction

The evolution of midlatitude weather is regulated predominantly by Rossby waves, the jet stream, and the life cycles of cyclones and anticyclones. Rossby waves propagate along typically zonal bands of strong meridional potential vorticity (PV) gradients, known as waveguides that co-align with the jet streams (Hoskins and Ambrizzi, 1993; Schwierz et al., 2004). The evolution of Rossby waves along the waveguides and the associated PV anomalies interact with the surface baroclinicity and modulate the genesis and evolution of low and high-pressure systems (Hovmöller, 1949; Hoskins et al., 1985) and extreme weather events in the extratropics (Wirth and Eichhorn, 2014; Röthlisberger et al., 2016a; Fragkoulidis et al., 2018; Röthlisberger et al., 2019).

Even though the dynamics of upper-level extratropical flows is mostly adiabatic and has comparatively high predictability (Cressman, 1948; Grazzini and Vitart, 2015; Quinting and Vitart, 2019), the upper-level flow is also substantially modified by strong diabatically induced divergent outflows, potentially leading to strongly amplified and persistent wave perturbations that favor surface weather extremes (Screen and Simmonds, 2014; Galfi and Messori, 2023). Research in the last decades has placed considerable focus on understanding the influence of diabatic processes in shaping the dynamics of upper-level flows and extratropical cyclones as reviewed in detail by Wernli and Gray (2024, in particular their Sect. 4.1.6 and 5.5). Early foundational studies (e.g., Atlas, 1987; Hoskins and Berrisford, 1988; Davis et al., 1993) discussed the potential influence of latent heat release on the intensity of upper-level ridges downstream of cyclones. Later studies built on these findings and analyzed the initiation and modulation of midlatitude Rossby wave patterns by diabatic outflows of warm conveyor belts (WCBs, Massacand et al. 2001; Röthlisberger et al. 2018) and transitioning tropical cyclones (Grams and Archambault, 2016; Riboldi et al., 2018). Other studies emphasized the contributions of WCBs to midlatitude forecast uncertainty (Rodwell et al., 2013; Gray et al., 2014; Martínez-Alvarado et al., 2016; Grams et al., 2018; Sánchez et al., 2020; Pickl et al., 2022). The interaction of WCB outflows with upper-level Rossby waves constitutes the theme of this study, which, for the first time, aims to systematically quantify WCB-waveguide interactions from a climatological perspective.

WCBs are coherent moist airstreams in extratropical cyclones, characterized by intense cross-isentropic ascent from the boundary layer to the upper troposphere and the subsequent divergent outflow in the region of the waveguide (Browning and Roberts, 1994; Wernli, 1997; Eckhardt et al., 2004). Along their ascent, they experience a substantial increase in potential temperature and decrease in specific humidity, leading to cloud formation and often intense surface precipitation (Wernli and Davies, 1997; Joos and Wernli, 2012; Madonna et al., 2014). From the potential vorticity (PV) perspective (Hoskins et al., 1985), the latent heat release leads to diabatic PV generation in the lower and PV destruction in the upper troposphere, respectively. Consequently, the net change in PV along WCB trajectories is nearly zero, with the average PV of the outflow essentially equal to that of the inflow (Wernli and Davies, 1997; Madonna et al., 2014; Methven, 2015). However, the cross-isentropic mass transport allows the outflow to reach high isentropic levels, resulting in anomalously low PV in WCB outflows relative to their surroundings. WCB outflows, therefore, often signify intense negative upper-level PV anomalies, with a strong potential to influence the downstream flow (Schemm et al., 2013; Madonna et al., 2014).

An essential motivation for this study is the fact that past studies reported different synoptic-scale flow structures for WCB outflows and their interaction with the upper-level waveguide. These structures include ridges, blocks, and tropospheric cutoffs,




as summarized in the following. The low-PV WCB outflow often enters and expands a pre-existing downstream ridge (Pomroy and Thorpe, 2000; Schemm et al., 2013). Through ridge amplification or ridge building, the diabatic WCB outflow also deflects the waveguide poleward and elevates the tropopause (Methven, 2015; Saffin et al., 2021). These synoptic flow conditions, induced by the diabatic divergent outflow, can serve as dynamical precursors of high-impact weather events downstream (Massacand et al., 2001; Martius et al., 2008) and initiate the downstream development of baroclinic Rossby wave packets (Grams and Archambault, 2016; Röthlisberger et al., 2018). WCB outflows can also play a crucial role in the formation, amplification, or maintenance of atmospheric blocks, influencing their intensity and lifespan (Pfahl et al., 2015; Steinfeld and Pfahl, 2019; Steinfeld et al., 2020; Kautz et al., 2022). The study by Steinfeld and Pfahl (2019) highlighted the significance of latent heating during the onset of blocking events and found that over 50% of the air parcels that contribute to a block experienced median heating of 12.5 K in the preceding seven days. Persistent flow features like blocks can contribute to extreme weather, such as heat waves (Pfahl et al., 2015; Zschenderlein et al., 2020) and cold spells (Sillmann et al., 2011). In addition to WCB interactions with ridges and blocks, it was reported that WCB outflows can also play a role in the formation of Arctic polar anticyclones (tropospheric PV cutoffs), which in turn affect the sea-ice variability over the Arctic (Wernli and Papritz, 2018). And finally, it is important to note that, however, some studies mentioned that strong diabatic outflow does not always lead to strong ridge building and, in cases with a particularly strong jet, can be rapidly advected downstream without disturbing the waveguide (Riboldi et al., 2018). This brief summary signifies that the impact of WCBs on Rossby wave dynamics varies greatly, and the nature of interaction significantly influences local and downstream weather conditions. Nevertheless, a comprehensive climatological analysis that systematically examines all dimensions of WCB-waveguide interactions is still missing.

To address the aforementioned open aspects regarding the WCB-waveguide interactions, the present study develops a method to systematically characterize these interactions to establish a global climatology and investigate their dynamics. From the above summary, it can be inferred that it is meaningful to consider four types of interactions that the WCB outflow can have with the upper-level waveguide: (a) weak or no interaction, where the waveguide remains fairly unperturbed, (b) formation or amplification of an upper-level ridge, (c) formation or amplification of an atmospheric block, or (d) formation of a tropospheric PV cutoff. This study aims to objectively identify and characterize these four interaction types, whereby the study seeks to answer the following research questions:

(i) What is the relative frequency of the four interaction types, and where do they occur relative to the climatological waveguide?

(ii) How do the WCB characteristics differ between the interaction types?

(iii) How do the ambient flow structures differ between the interaction types?

(iv) Is there seasonal and/or hemispheric variability in WCB-waveguide interaction types?

Accordingly, this paper is organized as follows. Section 2 discusses data and methods used to identify the flow features and categorize the WCB-waveguide interactions. The approach is then applied to four cases from the literature to illustrate the usefulness of the developed method. Section 3 presents the climatological frequencies of these interaction types, also considering





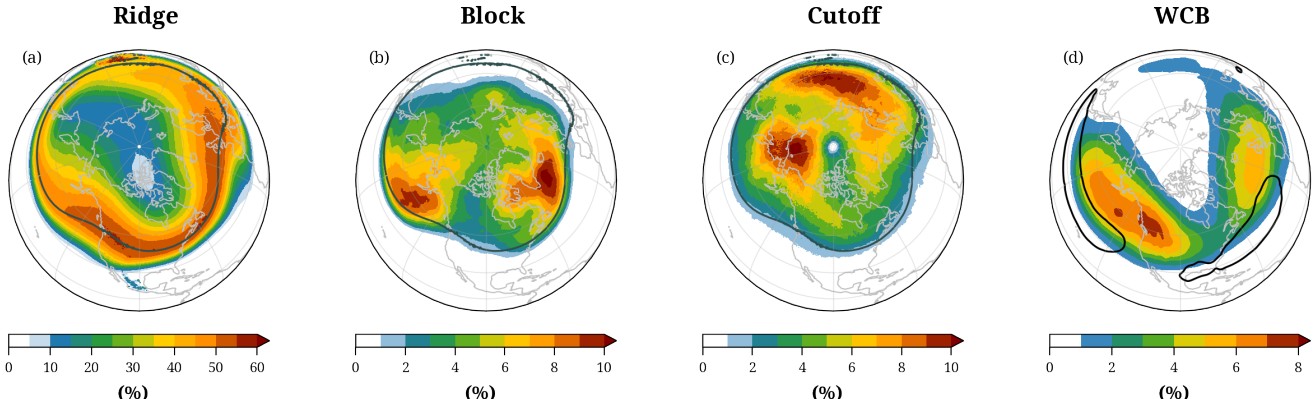

**Figure 1.** The climatological frequency of occurrence of weather features in DJF: **(a)** ridges, **(b)** blocks, **(c)** tropospheric cutoffs, and **(d)** WCBs at end-of-ascent. The black contours illustrate in (a-c) the climatological mean waveguide (i.e. the 2 PVU contour averaged for the four isentropes selected for DJF) and in (d) the 95th percentile of the WCB start-of-ascent frequency. Note the different contour intervals.

their variations across seasons and hemispheres. The properties of WCB trajectories involved in the distinct interactions are analyzed in Sect. 4. The characteristics of ambient synoptic flow conditions for various interactions that occurred during boreal winter are systematically considered in a composite analysis in Sect. 5. Finally, a summary of the key results and an outlook for future study are given in Sect. 6.

## 2 Data and methods

The study is based on 43 years (1980-2022) of ERA5 reanalysis data (Hersbach et al., 2020) developed by the European Center for Medium-Range Weather Forecasts (ECMWF). The hourly model-level data is interpolated from the original T639 spectral resolution to a 0.5° x 0.5° horizontal grid. The ERA5 dataset is used to compute various features and variables that represent the ambient upper-level flow situation and synoptic-scale features, as explained in the following subsections.

### 2.1 Weather system identification and calculation of other variables

The proposed classification of WCB-waveguide interactions into the four types (i) no or weak interaction, (ii) ridge, (iii) block, and (iv) tropospheric cutoff interactions, requires the identification of WCBs and of the potentially involved upper-level weather systems (ridges, blocks, and tropospheric cutoffs). These weather systems are computed as two-dimensional features based on isentropic or vertically averaged fields of absolute PV or PV anomalies (Sprenger et al., 2017).

For the identification of ridges, PV anomalies are calculated on isentropic levels, from 305 to 350 K (at 5 K intervals), as deviations from the 15-day running mean. Subsequently, the regions with PV anomalies less than $-1$ PVU (potential vorticity unit, $1$ PVU $= 10^{-6}$ m$^2$ s$^{-1}$ K kg$^{-1}$) in the Northern Hemisphere (NH) and with absolute PV values less than $2$ PVU are classified as upper-level tropospheric ridges on the considered isentropes. Upper-level blocks are identified as regions with



a vertically averaged negative PV anomaly (between 150 and 500 hPa, anomalies relative to the seasonal mean) that exceeds $-1.3$ PVU in the NH and persists for a minimum of 5 days (Schwierz et al., 2004; Croci-Maspoli et al., 2007). Even though identified from a two-dimensional field, the blocking region is considered valid on all isentropes from 305 to 350 K. Lastly, tropospheric PV cutoffs are detected on isentropic levels as isolated regions of PV values less than 2 PVU enclosed within the stratosphere (PV > 2 PVU) in the NH (Wernli and Sprenger, 2007; Sprenger et al., 2017). The same logic is applied in the

Southern Hemisphere (SH) with suitably adapted PV threshold conditions.

    With the approach described above, the weather features are available on isentropes from 305 to 350 K, every 5 K. However, the algorithm used for the WCB-waveguide interaction classification uses four specific levels that vary by month, as explained in the following subsection. It is important to mention that even on a given isentrope, the weather features may overlap; for instance, the region identified as a tropospheric cutoff can also be part of a block and/or a ridge (Fig. 2b below). This will have

implications for classifying and attributing interaction types (see Sect. 2.3).

    For calculating the climatological frequency of the three weather features, a feature is considered to exist at a specific location and time if it is present at any of the selected isentropes. In DJF, ridges are primarily identified near the mid-latitude waveguide (around 35–65°N), with a gradual frequency decrease towards the pole and a steep decrease on the equatorward side of the waveguide (Fig. 1a). Frequency maxima exceeding 50% span from the eastern North Pacific to western Europe.

In Fig. 1, the climatological mean waveguide is shown as the climatological mean 2 PVU contour of the PV averaged over the selected isentropic levels. The climatological frequency of blocks (Fig. 1b) agrees well with previous studies that used the same identification method but other reanalyses (Croci-Maspoli et al., 2007; Sprenger et al., 2017), with two maxima exceeding 10% over the western North Pacific, south of Alaska, and over the North Atlantic, south of Greenland, both at $50 - 60°$N. The tropospheric cutoff climatology (Fig. 1c) exhibits two maxima (again exceeding values of 10%): one over the

western Arctic ocean and the Chukotka peninsula at $60 - 80°$N and the other over Central Asia at about $60°$N. The cutoff climatology is comparable to the superposition of the tropospheric cutoff climatologies at 300–330 K by Wernli and Sprenger (2007, their Fig. 5). Blocks and atmospheric cutoffs occur almost exclusively poleward of the climatological waveguide. The primary occurrence regions of these features change considerably across seasons in the NH (Fig.S2) and comparably less in the SH (Fig.S3).

Another variable, which we will use in the last part of this study to quantify synoptic-scale activity, is the eddy kinetic energy (EKE), calculated as $\mathrm{EKE} = 1/2(u'^2 + v'^2)$, where $u'$ and $v'$ are the 10-day high-pass filtered zonal and meridional wind components at 250 hPa.

## 2.2 WCB identification

The Lagrangian analysis of the air parcels that constitute the different upper-level flow features provides comprehensive in-

formation about their origin and evolution. In this study, we focus on the air parcels near the waveguide, that ascended to the upper troposphere as part of a WCB in the previous five days. To identify WCB outflow air parcels in the upper troposphere, 120-h backward trajectories are computed from the vicinity of the waveguide, using the Lagrangian analysis tool LAGRANTO (Wernli and Davies, 1997; Sprenger and Wernli, 2015) and ERA5 three-dimensional wind fields. Aside from the trajectory



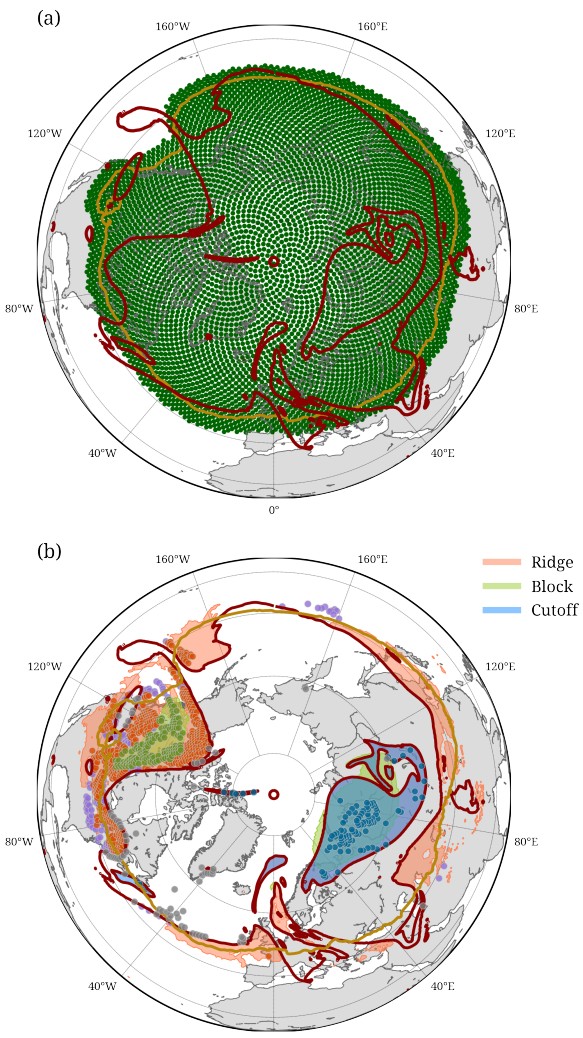

**Figure 2.** Example illustrating the Lagrangian analysis method to identify the four types of WCB-waveguide interactions at 00 UTC on 03 January 2016. The red and yellow contours represent the instantaneous and 30-day running mean waveguides (2 PVU isolines on 315 K), respectively. Green dots in **(a)** show the selected starting points on this isentrope for the five-day backward trajectory calculation. In **(b)**, regions identified as ridges, blocks, and tropospheric cutoffs are indicated with orange, light green, and blue shading, respectively. Dots in (b) show starting points of backward trajectories that satisfy the WCB ascent criterion, with the color indicating the interaction type: no-interaction (purple), ridge (orange), block (green), and cutoff (blue). Grey dots indicate stratospheric WCB air parcels, which are not included in the analysis.

position (longitude, latitude, pressure), variables such as potential temperature, PV, and specific humidity are also traced along
the trajectories. The starting regions for the backward trajectories are chosen such that they capture the regions where the WCB outflows are close to the Rossby waveguide, defined as 2 PVU isoline on isentropes (Martínez-Alvarado et al., 2016; Röth-



lisberger et al., 2016b; Wirth et al., 2018). Therefore, the vertical levels of the starting points depend on the climatologically preferred outflow levels reached by WCB trajectories. More specifically, the following criteria served to define the 6-hourly varying starting region (horizontal and vertical) for calculating the backward trajectories.

For the vertical refinement of the starting points, the monthly distribution of the isentropic level of the WCB outflows is analyzed separately for both hemispheres. For every WCB trajectory identified in the ERA5-based WCB climatology of Heitmann et al. (2024), the outflow isentropic level when the minimum pressure is attained along the WCB ascent is identified. As inferred from these climatological monthly distributions (Fig. S1), the WCB outflow varies seasonally in both hemispheres, with outflows reaching higher isentropes in summer (on average 335 K) than in winter (315 K). For our analysis, the vertical

levels for starting the backward trajectories in each month and hemisphere are determined by the median isentropic level ($\theta^*$, Table. S1) reached by the WCB outflows, and two isentropes below and one above this level: $\theta^* - 10\,\mathrm{K}$, $\theta^* - 5\,\mathrm{K}$, and $\theta^* + 5\,\mathrm{K}$. Using these four isentropic starting levels enables us to capture most of the WCBs along the 5-day backward trajectories.

For the horizontal refinement of the starting points, we use an approach that can cope with the complex geographical variability of the waveguide and Rossby wave patterns. First, the mean waveguide, defined as the 30-day running mean 2 PVU

isoline on the $\theta^*$ level, is identified (Fig. 2, yellow contour). This waveguide retains variability on scales larger than the synoptic scale and is, due to the time averaging, much smoother than the instantaneous waveguide, defined as the instantaneous 2 PVU isoline on the same isentrope (Fig. 1, red contour). The instantaneous waveguide exhibits large deviations from the mean waveguide, associated with troughs and ridges, and stratospheric and tropospheric PV streamers and cutoffs (Wernli and Sprenger, 2007). Regions where the instantaneous waveguide is poleward of the mean waveguide have negative PV anomalies,

and these regions are primary candidates for being associated with WCB outflows. Consequently, we chose the starting points of the backward trajectories poleward of the mean waveguide on the four selected isentropes, with a 5° equatorward buffer (dark green dots in Fig. 2a). The buffer serves to also capture the WCB outflow that does not strongly perturb the waveguide and is mainly advected downstream. The 120-hour backward trajectories are started every six hours during the period 1980–2022 on an equidistant grid ($\Delta x = 30\,\mathrm{km}$) in the horizontal region and on the monthly varying isentropes as described above.

To identify the WCB trajectories, the criterion of ascent of 600 hPa within a period of 48 h (Wernli and Davies, 1997; Madonna et al., 2014) is applied to any 48-h segment along the five-day backward trajectories. The colored dots in Fig. 2b mark the starting points of the five-day backward trajectories that fulfill the WCB criterion. These trajectories are referred to in the following as WCB trajectories. Note that they extend over five days, and the WCB ascent can occur during any two-day time window during this five-day period. All other trajectories, i.e., those that do not meet the WCB criterion, are disregarded

for further analysis in this study. For each WCB trajectory, we identify four distinct time instances (Fig. 3) as follows:

   i) Interaction: the starting point of the WCB backward trajectory is referred to as the 'point of interaction' of that trajectory (green dot in Fig. 3). This corresponds to the colored dots in Fig. 2b.

   ii) End of ascent: the first time (orange dot in Fig. 3), looking backward in time, that satisfies the WCB ascent criterion in the previous 48-hour time window (red box). This marks the time when the WCB ends its ascent, and we consider this

to be time zero for the WCB outflow.



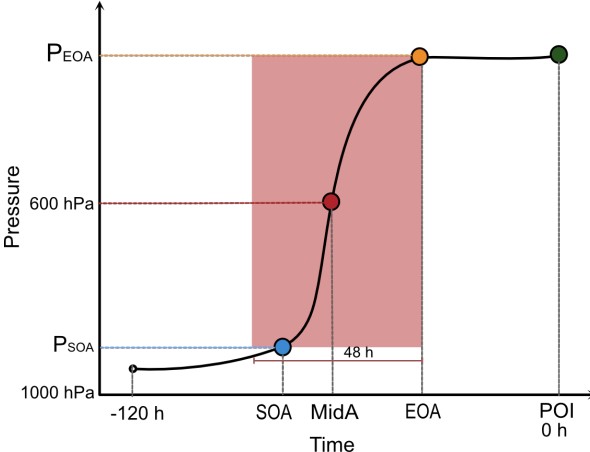

**Figure 3.** Schematic to introduce distinct WCB trajectory phases for a single five-day backward trajectory that fulfills the WCB criterion during the 48-hour time window indicated by the red box. The green dot denotes the starting point of the backward trajectory, which is also referred to as the 'point of interaction' (POI) of the WCB with the waveguide. The other dots denote the end-of-ascent (EOA, orange), the mid-ascent (MidA, red), which is when the trajectory crosses the 600-hPa level, and the start-of-ascent (SOA, blue).

iii) Mid-ascent: the first time (red dot), again looking backward in time, when the trajectory's pressure is equal to or less than 600 hPa.

iv) Start of ascent: the first time (blue dot), again looking backward in time, within the 48 h window from the end-of-ascent, when the pressure difference relative to end-of-ascent exceeds 600 hPa.

By design, the interaction point and the end-of-ascent can be identical. We refer to the time difference between the interaction and the end-of-ascent as the age of the WCB outflow at the interaction point, with a smaller age representing a fresh outflow and a larger age an old outflow. As shown later, the age of the outflow can vary strongly between different WCB-waveguide interaction events. The definitions of start-of-ascent, mid-ascent, and end-of-ascent in this study are in qualitative agreement with the concepts of WCB inflow, ascent, and outflow in previous climatological studies (Stohl, 2001; Eckhardt et al., 2004;

Madonna et al., 2014; Heitmann et al., 2024). The main regions of end-of-ascent in our climatology (Fig. 1d) are very similar to those of the WCB outflows in Heitmann et al. (2024, their Fig. 2e) and of the WCB trajectory positions after 48 h in Madonna et al. (2014, their Fig. 4f), with maxima over the central and eastern parts of the North Pacific and North Atlantic, respectively. The mid-ascent regions are comparable to the ascent climatology by Heitmann et al. (2024, their Fig. 2c). And the start-of-ascent regions (Fig. 1d, black contours) correspond with the inflow regions in Heitmann et al. (2024, their Fig. 2a) and

the starting regions of forward WCB trajectories in the climatology by Madonna et al. (2014, their Fig. 4d). These favorable comparisons reveal the robustness of our method, which is based on backward trajectory calculations from near the waveguide, in effectively identifying the majority of WCBs.





## 2.3 Attribution of WCB-waveguide interaction type

As the essential last step of our algorithm, for each WCB trajectory, the category of interaction is determined according to
the upper-level weather features (if any) that occur at the geographical location and on the isentropic level of the trajectory's
starting point (see colored dots in Fig. 2b and green dot in the schematic Fig. 3). If the starting point does not intersect with
any feature, the WCB trajectory is classified as a 'no interaction' type. A hierarchy from 'no interaction' to 'ridge' to 'block'
to 'cutoff' is followed when classifying the trajectories. For instance, if the starting point of a WCB trajectory is collocated
with both a ridge and a cutoff (i.e., if these features overlap), then, following this hierarchy, this trajectory will be classified
as a cutoff interaction type. In this way, each trajectory is attributed to only one interaction type. This hierarchy reflects the
increasing intensity of the waveguide disturbance associated with each interaction type: cutoff interactions are considered the
most intense as they involve non-linear Rossby wave breaking, while the no-interaction type represents events with only weak
or no waveguide disturbance. In rare cases, WCB trajectories occur with starting points in the stratosphere, i.e., with PV values
greater than 2 PVU in the NH (grey dots in Fig. 2b). This corresponds to events of troposphere-to-stratosphere transport in the
WCB outflow (Wernli and Bourqui, 2002), which are, however, excluded from the analysis in this study even if previously
attributed to one of the interaction categories.

## 2.4 Interaction examples

Before the method introduced above is applied climatologically, we revisit four well-documented WCB case studies from the
literature and visualize the classification of the WCB-waveguide interaction for these examples.
Figure 4a shows an event of rapid frontal wave cyclogenesis in the North Atlantic with an intense WCB and the formation
of a coherent PV tower in the mature stage of the cyclone (Wernli, 1997; Rossa et al., 2000; Heitmann et al., 2024). At the time
shown, the mature cyclone is located near Iceland, and the WCB outflow fills part of the large downstream ridge over northern
Europe, in particular its westward extension towards Greenland. These WCB air parcels are classified as having ridge-type
interaction (orange dots), with most of the air parcels very close to the cyclone center, consistent with the previous findings of
a prominent cyclonic WCB branch associated with this cyclone (Heitmann et al., 2024). The WCB contribution to the ridge
amplification is consistent with the previous studies that emphasized the strong negative PV anomaly in its outflow. Further
east, over Russia, the algorithm identifies another ridge with only a few WCB interactions, illustrating the well-known fact that
the diabatic contribution to upper-level ridge-building is highly case-dependent.

Another case of a strong WCB outflow with a ridge interaction over western Europe is shown in Fig. 4b, previously investi-
gated by Joos and Wernli (2012). Compared to the previous case, the associated cyclone is weaker, and at 00 UTC 31 January
2009, the narrow ridge extends meridionally from the western Mediterranean to northern Scandinavia. WCB air parcels fill
large parts of the ridge, particularly its northernmost extension. Few WCB air parcels over northern Africa were classified as
non-interacting (purple dots).

The following case (Fig. 4c) depicts a tropospheric cutoff interaction over northern Russia. A time series of this case (not
shown) indicates that the WCB outflow of cyclones over the eastern North Atlantic first intensified a ridge, which merged with





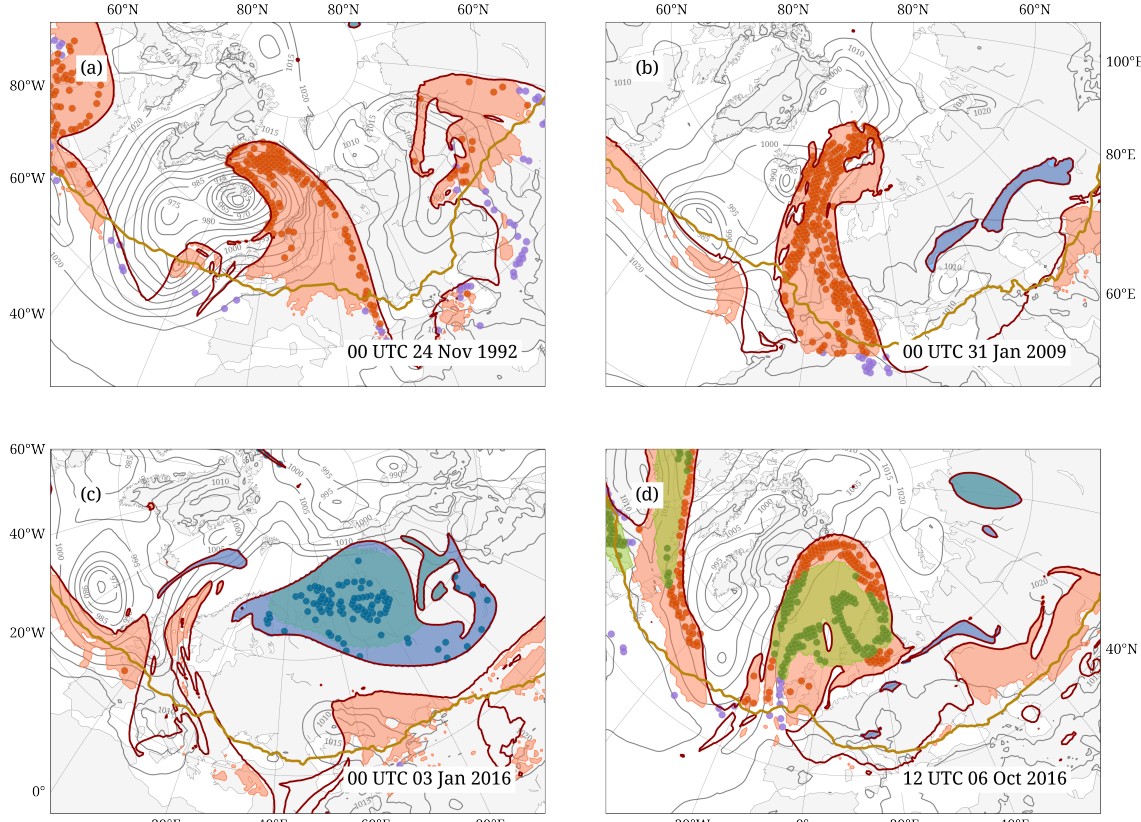

**Figure 4.** Four case studies illustrating WCB-waveguide interaction events previously discussed in the literature, at **(a)** 00 UTC on 24 November 1992, **(b)** 00 UTC on 31 January 2009, **(c)** 00 UTC on 03 January 2016, and **(d)** 12 UTC on 06 October 2016. The elements shown are the same as in Fig. 2b, except that here, stratospheric WCB air parcels are not shown.

an existing block and eventually led to the formation of the tropospheric cutoff. The WCB outflow and the induced cutoff-high contributed to strong poleward transport along the western flank of the cutoff and exceptional Arctic warming in the winter of 2015/16 (Binder et al., 2017; Kim et al., 2017). The tropospheric cutoff shown in Fig. 4c coincides partially with a block feature; however, following the hierarchy, the WCB air parcels are classified in this case as cutoff interaction (blue dots).

While previous studies highlighted the role of WCBs in transporting warm moist air into the Arctic, their interaction with the waveguide leading to the eventual formation of tropospheric cutoffs was not explicitly identified in their analyses.

The last example depicts the atmospheric block Thor that occurred in October 2016 east of Greenland during the North Atlantic Waveguide and Downstream Impact Experiment (NAWDEX; Schäfler et al., 2018). The block is situated within a larger ridge that extends further poleward (Fig. 4d). The algorithm identifies different types of interaction for the associated

WCB air parcels. The ones collocated with the block are of type block interaction (green dots), the ones further poleward are of type ridge interaction (orange dots), and very few no-interaction WCB air parcels (purple dots) are found south of the ridge.





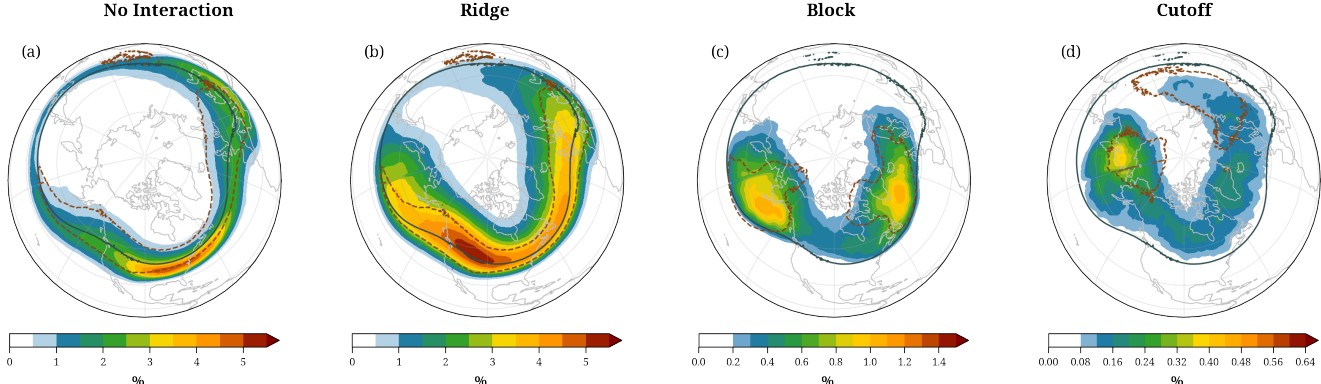

**Figure 5.** The climatological frequency of occurrence of WCB trajectories at their point of interaction (colors, in %), in DJF 1980–2022, for the four interaction types **(a)** no-interaction, **(b)** ridge, **(c)** block, and **(a)** cutoff interactions. Note the different contour values in the four panels. The black contours illustrate the climatological mean waveguide in DJF (2 PVU), and the dashed red contours show the 95th percentile regions of the associated weather features (a,b) ridges, (c) blocks, and (d) cutoffs.

The WCB air parcels that are part of the overlapping region of the block and ridge are classified only as block interaction, following the hierarchy rule mentioned earlier. The contribution of the WCB outflow to sustaining this block was demonstrated by Steinfeld et al. (2020) through numerical experiments. Our method further confirms the interaction of the WCB with the block, validating its role in block maintenance.

Overall, the four examples served to illustrate the functioning of our classification approach in strongly differing flow situations. This enables us to investigate the climatological characteristics of the four interaction types.

## 3 Geographical distributions of interaction types

The interaction type attribution is conducted for all WCB backward trajectories calculated during the 43 years from 1980 to 2022. The climatological analysis for DJF reveals that WCB trajectories are most often classified as ridge interacting type (54.0%), followed by no interaction (25.5%), while block (8.9%) and cutoff (3.6%) interactions are less common. Each interaction type tends to occur in specific preferred regions.

To comprehend the geographical occurrence, the horizontal positions of the WCB trajectories are gridded onto a regular grid with 0.5° horizontal resolution, every six hours, separately for the different ascent phases (see Fig. 3). This gridding yields binary fields with a value of 1 if the corresponding grid cell contains at least one WCB trajectory, irrespective of the vertical level (and 0 otherwise). The climatological frequency fields are then calculated as averages of these six-hourly binary fields, yielding the percentage of six-hourly time steps with the presence of WCB trajectories. The fields are smoothened using a two-dimensional Gaussian filter. The following subsections discuss the climatologies of WCB points-of-interactions (Sect. 3.1) and other phases (Sect. 3.2) for the four interaction types.





## 3.1 Point of interaction

The climatological frequency fields of WCBs at the point of interaction correspond to the probability of having a WCB outflow
at that location on at least one of the four considered isentropic levels, which ascended from the lower troposphere within the
past five days. These frequencies vary strongly across the four types of interaction (Fig. 5). The highest frequencies exceeding
5% occur for the ridge interaction type over western North America (Fig. 5b). The highest frequencies of no interaction (4.5%)
are found over the eastern US (Fig. 5a), of block interactions (exceeding 1%) in regions south of Alaska and south of Greenland
(Fig. 5c), and of cutoff interactions (about 0.4%) over eastern Siberia (Fig. 5d). The differences in frequencies can partly be
explained by the differing frequencies of the associated weather systems, with ridges being much more frequent than blocks
and cutoffs (Fig. 1).

The primary regions of the points of interaction for the no-interaction type (Fig. 5a) closely follow the climatological waveg-
uide, reflecting the advection of WCB air parcels along the waveguide that do not engage in the formation of a ridge, block,
or cutoff. The no-interaction frequency fields show a continuous band extending from the central North Pacific over North
America (with the previously mentioned frequency maximum of 4.5%) and the North Atlantic towards southern Europe (with a
secondary maximum of 3% over the eastern Mediterranean) and the Caspian Sea. As expected, it can be seen that no-interaction
primarily occurs equatorward of the climatological waveguide.

Ridge interactions (Fig. 5b) also tend to follow the climatological waveguide, with the majority of interactions along and
slightly poleward of the waveguide. High values exceeding 3% occur in a band extending from the dateline across North Amer-
ica to central Europe. The regions with frequent ridge interactions of WCBs align qualitatively with the primary climatological
ridge regions (Fig. 1a). However, it is important to notice that ridge frequencies across North America and the North Atlantic
exceed 50%, whereas WCBs with ridge interactions occur more than 10 times less frequently. This discrepancy does not imply
that only a few ridges are associated with WCB interactions, but rather, these WCB interactions typically do not occur in the
entire area identified as a ridge. This was shown for two case study examples in Fig. 4a,b, where WCB interaction points filled
about 20 and 50% of the respective ridge areas.

The locations of block and cutoff interactions (Fig. 5c,d) are also influenced by the climatological distributions of blocks and
cutoffs. In particular, for blocks, their frequency maxima over both ocean basins (Fig. 1b) correspond well with the maxima of
WCB block interaction points. Also, since block features are identified almost exclusively poleward of the mean waveguide, the
same applies, by design, for block interaction points. For cutoffs, the link between their climatological occurrence (Fig. 1c) and
the WCB cutoff interactions is less strong. The region with maximum cutoff interactions over the Kamchatka Peninsula, the
Bering Sea, and eastern Siberia (Fig. 5c), is slightly shifted equatorward of the cutoff frequency maximum over eastern Siberia
(Fig. 1c). Interestingly, the second region with a cutoff frequency maximum over central Asia is related to almost no WCB
interactions, which is most likely because of the very low WCB outflow frequency in this region and upstream of it (Fig. 1d).
Instead, cutoff interactions also occur relatively frequently in a band at about 60°N that extends from Alaska to Scandinavia.
Block and cutoff interaction frequency maxima are roughly 10-20 times smaller than the corresponding frequency maxima
of block and cutoff features, respectively, again for the two reasons that (i) not every block/cutoff is associated with a WCB




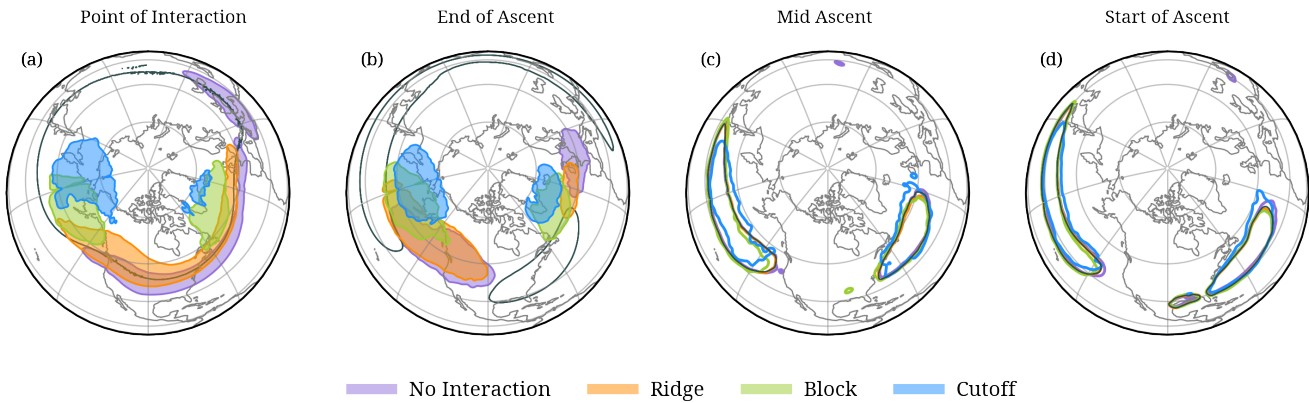

**Figure 6.** The main climatological frequency maxima (95th percentile regions) of WCB trajectories for: no-interaction (purple shading and contours denoting the regions where the interaction type frequency exceeds the 95th percentile), ridge interaction (orange), block interaction (green), and cutoff interaction (blue). The maxima are shown at the four trajectory phases **(a)** point of interaction, **(b)** end-of-ascent, **(c)** mid-ascent, and **(d)** start-of-ascent. In (c,d) shading is omitted to enhance readability. The black contours illustrate in (a) the climatological mean waveguide, in (b) the climatological jet stream identified by a wind speed of $30\,\mathrm{m\,s^{-1}}$, where both are averaged over the four isentropic levels considered, and in (c) and (d) the 95th percentile regions of total WCB mid-ascent and start-of-ascent, respectively. Note the different contour intervals.

interaction, and (ii) WCB interactions typically do not fill the entire regions identified as blocks/cutoffs, as illustrated in the examples in Fig. 4c,d.

## 3.2 Comparison of WCB ascent regions between interaction types

In the previous section, we considered the climatological frequency distribution of the four interaction types at the point of interaction. The location of the maxima of these distributions differs substantially between the interaction types. In fact, the maxima differ in terms of longitude, and one can assume that the points of interaction are located less downstream relative to the WCB ascent for high intensity interaction. This assumption will be further validated in this subsection. In addition, Fig. 5 showed a clear increase in the latitude of the points of interaction with increasing interaction intensity: the latitude of maxima in the interaction point distributions increases from about 37°N (no interaction) to 45°N (ridges) to 55°N (blocks) to 62°N (cutoffs). This can be seen again clearly in Fig. 6a, which shows the regions with the highest 5% of the interaction point frequencies (95th percentile), but because of the lowest number of WCB trajectories for the cutoff type, the 95th percentile contours also appear slightly more noisy. We are now interested to see how these clear shifts in the distributions at the points of interaction translate into shifts in the distributions at earlier times of the WCBs. In particular, we are curious to identify potential differences in the inflow and ascent regions of WCBs in the four interaction types.

Figure 6b-d compares the maxima of the WCB frequency distributions at times denoted as end-of-ascent, mid-ascent, and start-of-ascent, respectively (see definitions in Fig. 3). It becomes obvious that the distributions for the four interaction types





differ much less in earlier WCB phases. At the end-of-ascent (Fig. 6b), the distributions differ in a similar way, but less strongly, than the distributions at the point of interaction (Fig. 6a). Again, with increasing interaction intensity, the maxima are shifted westward and poleward, both over the North Pacific and the North Atlantic. It is also apparent that, in both basins, maxima in end-of-ascent occur poleward and to the east of the climatological mean jets (Fig. 6b). The end-of-ascent regions of the no-interaction and ridge categories only differ negligibly in their latitudinal position (and, over the North Pacific, also

in terms of longitude; Fig. 6b). Comparing the point-of-interaction with the end-of-ascent regions, the no-interaction and ridge interaction categories exhibit a strong eastward advection (except for ridge interactions over the North Atlantic), with an additional equatorward shift for the no-interaction type (Fig. 6a,b). In contrast, the locations for block and cutoff interactions remain fairly similar to the ones for end-of-ascent. These marginal differences between the point-of-interaction and end-of-ascent distributions reveal the stationarity of the anticyclonic circulation associated with blocks and tropospheric cutoffs.

In stark contrast to the points of interaction and the end-of-ascent regions, the regions with maximum mid-ascent and start-of-ascent are surprisingly similar for the four interaction types (Fig. 6c,d). In fact, they all agree essentially with the climatological WCB distributions indicated by the black contours. Minor differences can be seen for the cutoff interaction type, with the main start-of-ascent region located slightly further east and north. This lack of preferred regions for start-of-ascent and mid-ascent suggests that the WCB-waveguide interaction type is largely independent of their starting and ascent location. Nevertheless, the

interaction types differ primarily in terms of the latitudinal position of the WCB outflow. The WCB outflow and its downstream evolution are most likely strongly influenced by the background flow, since the flow is predominantly adiabatic after the end-of-ascent. This aspect will be investigated further in Sect. 5 and in Sect. 4, where we consider additional properties of the WCB trajectories in the four categories. This will help us better understand how the properties of the WCB air parcels at the end-of-ascent and the ambient flow conditions together determine the interaction of the WCB outflow with the waveguide. But

since the discussion so far focused on one season and the NH only, the next subsection summarizes the results from similar investigations globally and year-round.

### 3.3 Other seasons and the Southern Hemisphere

The same analyses as discussed above for NH winter were performed for all seasons in both hemispheres. Figures corresponding to Figs. 5 and 6 for all seasons and both hemispheres can be found in the Supplement (Figs. S4, S5, S6, and S7). The

335 overall patterns of WCB interactions remain consistent across all seasons and hemispheres. The absolute frequency of different interaction types varies with seasonal changes in WCB activity, with the lowest frequencies in the summer. Nevertheless, the ridge interactions were most frequent, followed by no-interaction and the two types of most intense interactions, which are even rarer during other seasons.

Regardless of season or hemisphere, the relative positioning of interaction regions with respect to the climatological waveg-

340 uide remains consistent, with no-interaction events occurring equatorward, while ridge, block, and cutoff interactions are progressively shifted poleward. Seasonal variations primarily affect the longitudinal position of no-interaction and ridge interactions due to differences in the strength of the ambient westerlies and associated eastward advection. The locations of block and cutoff interactions are also influenced by seasonal shifts in the occurrence regions of the corresponding weather systems.



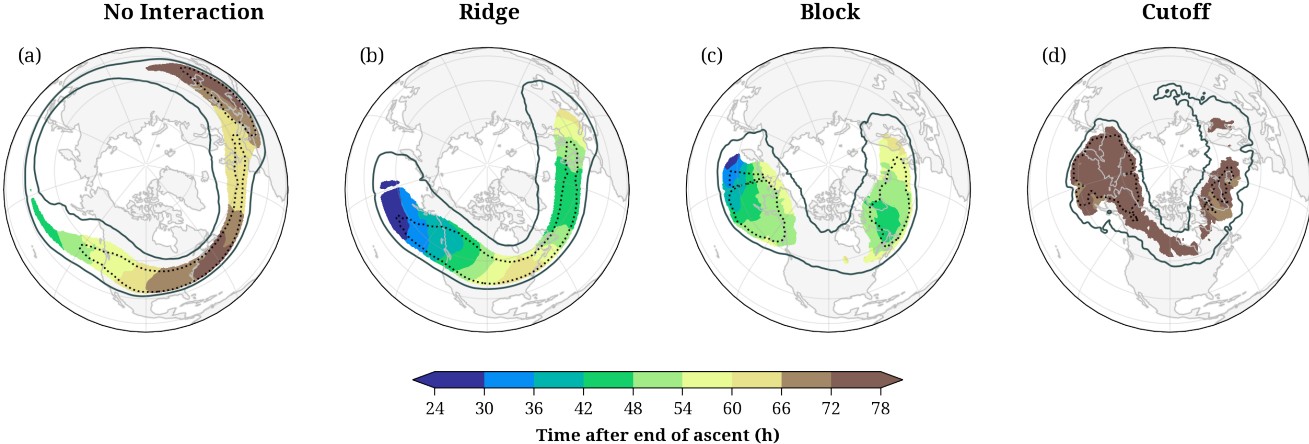

**Figure 7.** The climatological mean WCB outflow age at the point of interaction (colors, in h) for **(a)** no-interaction, **(b)** ridge, **(c)** block, and **(d)** cutoff interaction types in DJF. The black contours show the 80th (solid) and 95th (dotted) percentiles of WCB frequencies at the point of interaction for the respective interaction types.

In the SH, during JJA, about 63.6% of WCB outflows in SH interact with a ridge, 30.5% are non-interacting, and very few result in block (3.3%) and cutoff (2.6%) interactions. Here, the WCB point-of-interaction regions form a continuous and widespread pattern, spanning the 30°S–70°S latitudinal band in all seasons. Particularly, the ridge and no-interaction events extend across the entire mid-latitude band, with minimal seasonal variability. The block and cutoff interactions occur mainly over the ocean domains, with blocks being comparatively frequent during DJF and cutoffs during JJA (Fig. S5).

The poleward shift in the end-of-ascent regions of different interaction categories was also observed for the different seasons and the SH (Fig. S6 and S7). The seasonal change in the strength of westerlies affected the evolution from the end-of-ascent to the point-of-interaction, with the summer season having the least difference between these regions, even for the weaker interaction types. For all interaction types, the start-of-ascent and mid-ascent regions are almost identical for the four categories in all seasons and hemispheres. The start-of-ascent regions show seasonal changes inherent in WCBs, with these regions predominantly confined over the western ocean basins during summer.

## 4   Lagrangian properties of interaction types

This section analyzes a set of Lagrangian characteristics of WCB trajectories to investigate whether the trajectories associated with different interaction types exhibit distinct properties. The characteristics considered are the age of the outflow at the point of interaction and the evolution of pressure, potential temperature, humidity, and PV along the WCBs. This analysis will focus mainly on the period from the end-of-ascent to the point-of-interaction, because during this period the spatial evolution of the WCB trajectories differs most between the interaction types.





### 4.1 Outflow age

The average age of the WCB outflow at the point-of-interaction, i.e., the average time since the end-of-ascent, shows interesting differences between the interaction types (Fig. 7). A young outflow age (of less than one day) indicates that the interaction occurs soon after the end-of-ascent of the WCB trajectories and, therefore, also geographically close to the WCB ascent. In contrast, an older outflow age (e.g., 3 days or more) implies that interactions with the waveguide happen in later stages of the outflow and, given the strong winds near the waveguide, potentially far downstream from the WCB ascent.

WCBs of the no-interaction type have typical outflow ages older than two days (Fig. 7a). The youngest interaction points of this type occur over the central North Pacific (with an age of about 42 h), then the age increases downstream and exceeds 72 h near the frequency maximum of WCB no-interactions over the North American east coast. Further downstream, over the North Atlantic, additional WCB ascent reduces the outflow age there, before it increases again towards the Mediterranean. The outflow age distribution reveals that the peak region of no-interaction corresponds mainly to old outflow advected eastward from the western North Pacific. The longitudinal distribution of the outflow age for ridge interactions is very similar, albeit with about 18 h lower values (Fig. 7b). Again, the age of the outflow is youngest over the central North Pacific, it has a local minimum over the North Atlantic, and is oldest over the eastern US and southern Europe. The block interactions reveal a similar outflow age as ridge interactions, whereas cutoff interactions occur on average more than 72 h after the end-of-ascent (Fig. 7c,d).

When interpreting these results, one must keep in mind that Fig. 7 shows average climatological values, and there is most likely a large variability among WCBs of the same interaction type. Nevertheless, we found a systematic pattern that ridge interactions are the youngest, in particular over the North Pacific. Block interactions are only marginally older, whereas cutoff interactions and no-interactions occur 2-3 days after end-of-ascent. The outflow age varies also across seasons and hemispheres, but consistently, younger outflows contribute mostly to ridge and block interactions, while older outflows are linked often to cutoff or no interaction (Fig. S8, S9).

Following the outflow age distribution, a WCB trajectory can first be classified as having ridge interaction (i.e., the WCB trajectories contribute to the formation of a ridge shortly after end-of-ascent), but later, in case the trajectories leave the ridge and are advected along an intense jet downstream, they might change to no-interaction type. Alternatively, the ridge might get involved in a Rossby wave breaking, and the aged WCB outflow becomes part of a tropospheric cutoff and is, therefore, classified into the cutoff interaction type. With both scenarios, it is plausible that the outflow age is the oldest for the cutoff and no-interaction types. We can also interpret the results shown in Fig. 7 in the following way: since young WCB-waveguide interactions occur preferentially in ridges and blocks, these flow features "profit" most from the injection of low-PV air by WCBs. This finding is in line with the many studies emphasizing the role of WCBs for ridge building and the formation and maintenance of blocks (see, e.g., Sect. 5.5 in Wernli and Gray, 2024).

 

**Table 1.** The median values and standard deviations of start-of-ascent specific humidity (in $\mathrm{g\,kg^{-1}}$), integrated latent heating (in K), and the end-of-ascent isentropic level (in K), averaged over all northern hemisphere WCBs in DJF, separately for each interaction category.

| Interaction types | specific humidity $q_{SOA}$ | integrated heating $\Delta\theta$ | potential temperature $\theta_{EOA}$ |
|---|---|---|---|
| No interaction | 8.31 / 1.86 | 23.73 / 6.33 | 320.14 / 9.80 |
| Ridge | 8.68 / 1.82 | 24.45 / 6.44 | 320.41 / 10.27 |
| Block | 8.61 / 1.81 | 22.74 / 5.32 | 319.70 / 8.43 |
| Cutoff | 8.17 / 1.58 | 20.81 / 5.25 | 316.38 / 8.42 |

## 4.2 WCB characteristics

We now change the perspective and consider the statistics of several WCB characteristics in the four categories. Table 1 shows the median values and standard deviations for the four categories (averaged over all NH WCBs in DJF) of start-of-ascent specific humidity, integrated latent heating (i.e., the potential temperature difference between end-of-ascent and start-of-ascent), and the end-of-ascent isentropic level. In accordance with the previous finding that WCB inflow regions are very similar for the four interaction types, we find only minor differences in specific humidity in the WCB inflow. WCBs in the interaction categories ridge and block are slightly moister (by $0.3 - 0.5\,\mathrm{g\,kg^{-1}}$) than WCBs in the other categories. The integrated latent heating also differs only slightly. Consistent with the slightly higher inflow humidity, the averaged values are a bit more than 1 K higher for ridge and block interactions. The interaction categories do not exhibit a specific preference in isentropic levels, except for the cutoff interaction, which is observed to occur at lower levels. More important differences are identified in other WCB outflow characteristics, such as pressure and PV, as discussed in the following paragraphs.

The pressure level reached by the WCB air parcels at the end-of-ascent and point-of-interaction decreases with increasing interaction intensity, from the no-interaction to the cutoff interaction type (Fig. 8a). The pressure changes from the end-of-ascent to the point-of-interaction vary across the interaction types and occur over differing time periods in accordance with the differing outflow age associated with each interaction. The case-to-case variability is also significant and becomes more pronounced at the point-of-interaction than at the end-of-ascent.

The WCBs associated with weak interactions reach lower altitudes (i.e., higher pressure of, on average, 316 hPa) at their end-of-ascent and descend, within 36 h, to even higher pressure levels (366 hPa) at the point-of-interaction (Fig. 8a, purple). This pressure increase at the point-of-interaction is consistent with the equatorward shift along the sloping isentropes observed for the no-interaction type (Fig. 6a, purple). In contrast, cutoff interactions finish their ascent at a lower pressure of roughly 263 hPa, further ascending to 257 hPa until the interaction point, indicative of poleward advection along isentropes as they age (Fig. 8a, 6a, blue). WCBs with ridge and block interactions, however, maintain their post-ascent pressure levels, with medians of 307 hPa and 295 hPa, respectively, as these interactions typically have a young age of less than 20 h (Fig. 8a, orange



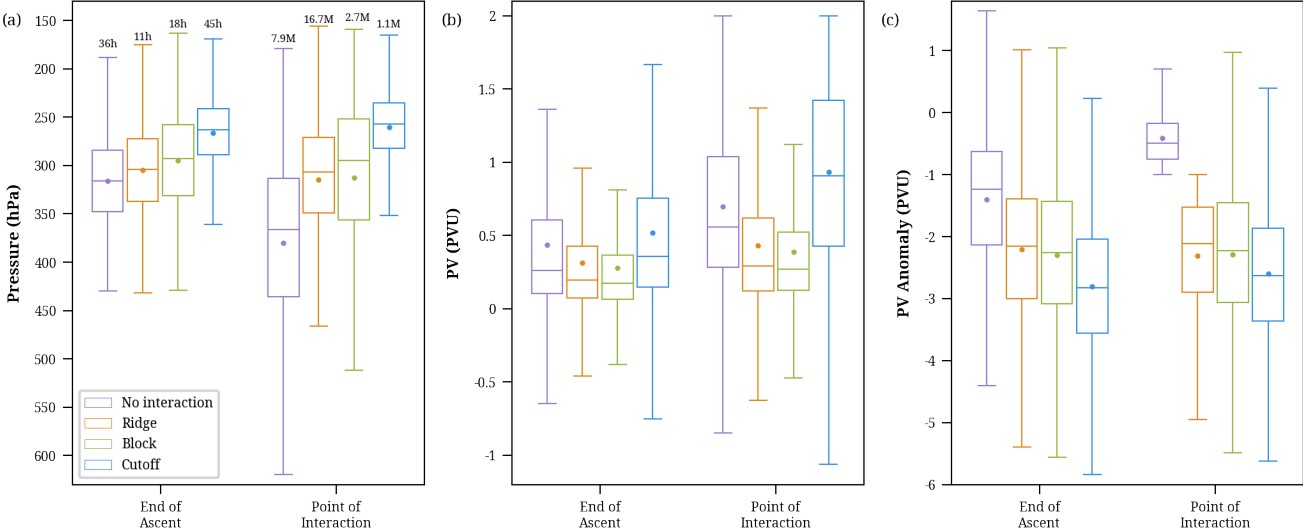

**Figure 8.** Characteristics of different WCB interaction types at the end-of-ascent and point-of-interaction in DJF. Shown are box plots of **(a)** pressure (hPa), **(b)** absolute PV (PVU), and **(c)** PV anomalies (PVU). Indicated are the median (horizontal line), the mean (circular dot), the interquartile range (box), and the whiskers extending to the 5th and 95th percentiles; outliers are not shown. The numbers in (a) above the bars indicate (left) the median outflow age and (right) the total number of trajectories.

and green). Hence, WCBs that reach the lowest pressure levels at the end-of-ascent are likely to experience the most intense interaction.

Regardless of the outflow pressure and consistent with earlier studies (Madonna et al., 2014), most WCBs at the end-of-ascent exhibit low PV values of about $0.1 - 0.7\,\text{PVU}$. These values decrease, on average, from no interaction to ridge to block interaction (Fig. 8b). However, WCBs associated with the most intense interaction in the cutoff type have the highest PV values, with a median of approximately $0.36\,\text{PVU}$ (Fig. 8b, blue). PV, like pressure, also increases as WCBs evolve from the end of their ascent to the point-of-interaction, particularly notable for no-interaction and cutoff interactions. This increase in PV most likely results from radiative processes and turbulent mixing near the jet.

The cutoff interaction retains the highest PV ($0.91\,\text{PVU}$) at the point-of-interaction, with the largest PV gain since the end-of-ascent. Higher PV values for cutoff interactions are also observed at the start-of-ascent (median $\sim 0.6\,\text{PVU}$, not shown), which might be attributed to the climatologically poleward shifted starting regions (Fig. 6d) compared to the other categories ($\sim 0.45\,\text{PVU}$). Interestingly, at both considered times PV values and their variability are smallest for WCBs in the block interaction type. Negative PV is found occasionally at both WCB phases in all categories.

Madonna et al. (2014, their Fig. 8) showed that the low PV values in WCB outflows correspond to strongly negative PV anomalies since these air parcels reach regions where PV is climatologically high. We find a large increase of this anomaly magnitude as the interaction intensity increases (Fig. 8c). The no-interaction WCBs possess the weakest (least negative) PV





anomaly of about $-1.2$ PVU at the end-of-ascent, which evolves to $-0.5$ PVU until the point-of-interaction. This reduction of negative PV anomalies in no-interaction WCBs can be attributed to the typically equatorward motion after end-of-ascent (Fig. 6a,b, purple regions). It be should be noted that the PV anomalies in the no-interaction type at the point-of-interaction are limited by the classification method, which specifies that the PV anomalies must be weaker than $-1$ PVU. In contrast, the cutoff interactions exhibit the strongest averaged PV anomaly at the end-of-ascent ($-2.8$ PVU) and almost retain this low value until the point-of-interaction ($-2.6$ PVU), 45 h later (Fig. 8c, blue). Since cutoff interactions typically occur far poleward (Fig. 5d), where the climatological PV is inherently large, the WCBs can attain stronger PV anomalies even when the PV values themselves are larger than for the other interaction categories. The ridge and block categories have comparable PV anomaly values at the end-of-ascent and point-of-interaction. Since these interactions happen soon after the ascent, the WCBs roughly maintain their anomalies until the point-of-interaction, with a median of $-2.1$ and $-2.2$ PVU, respectively (Fig. 8c, orange and green). The PV characteristics of the interaction types indicate that the latitudinal and vertical position of the outflow, and therefore the ambient PV climatology, are more relevant than the absolute PV value in the WCB outflow for explaining the resulting PV anomaly at the point-of-interaction.

In summary, the WCB outflows leading to weak interaction exhibit weaker negative PV anomalies and higher pressure values at the end-of-ascent and point-of-interactions, while those leading to intense interaction possess more pronounced negative PV anomalies at lower pressure values. Similar behaviour is observed for other seasons (Fig. S10, S11, S12) and also in the SH (Fig. S13).

## 5 Upper-level flow conditions and their time evolution

The results thus far have highlighted differences among the interaction types, particularly in terms of the outflow latitude and characteristics. In addition, the discussion has revealed the significant role of the upper-level flow in shaping the post-ascent evolution of WCB air parcels, thereby influencing the nature of their interaction with the waveguide. Building on this, the study now delves deeper into examining the upper-level synoptic conditions that favor each interaction type.

This analysis is only conducted for the two major NH ocean domains in DJF, which feature high WCB frequencies and distinct point-of-interaction and end-of-ascent patterns for all interaction categories. These regions are in the North Pacific ($150°$E to $170°$W, $20°$ to $60°$N) and in the North Atlantic ($80°$W to $40°$W, $20°$ to $60°$N) and they are analyzed separately since their background flow conditions are distinctly different, with a stronger and more zonal climatological jet with less variability over the North Pacific and a highly variable jet with comparatively lower climatological maxima over the North Atlantic.

At each 6-hourly time step in ERA5 during the period 1980–2022, the WCB trajectory positions are compiled separately for the different ascent phases onto a regular grid. For each phase, we then select the time steps with more than 200 air parcels in the corresponding phase and within the ocean domain for further analysis. With this criterion, we make sure that for the composites, we only consider time steps when a certain WCB phase occurs prominently in the domain. The percentage of air parcels contributing to different interaction types in each phase is evaluated to attribute the selected time step to a particular interaction type. Due to the unequal number of trajectories and occurrence frequencies of each interaction type, a subjective





**Figure 9.** Composites of ridge-interaction time steps for the four trajectory phases: point of interaction **(a,e)**, end-of-ascent **(b,f)**, mid-ascent **(c,g)**, and start-of-ascent **(d,h)**. (a-d) show PV at 320 K and geopotential height anomalies at 250 hPa (solid and dashed lines for positive and negative values, respectively). (e-h) show 500 hPa EKE anomalies and low-pass filtered zonal wind at 250 hPa (brown lines). The highlighted white-green contours in (a-h) show the 98th and 99th percentile of occurrence frequency of the ridge-interaction WCB air parcels in the corresponding phases. Only the WCB air parcels that ascend within the rectangle (c,g) are selected for the analysis. The number of time steps included in each composite and the corresponding percentage w.r.t. all DJF time steps in the considered 43-y period are given in each panel.





threshold is applied instead of a simple majority rule. If more than 40% of the WCB air parcels in the domain belong to the
no-interaction type in a specific ascent phase, then this time step is attributed to that phase and the no-interaction type. If not,
and more than 10% of air parcels belong to the cutoff-interaction type, the time step is assigned to that category. If neither
condition is met, thresholds of 15% and 65% are applied in a hierarchical order, respectively, for block and ridge interactions,
ensuring that each time step is attributed to only one interaction type per ascent phase. To calculate the final composites, this
attribution is repeated for all time steps and ascent phases in both ocean domains. If none of the conditions is met, the timestep
is not used for the analysis.

Despite a considerable case-to-case variability, the composites created using the selected time steps reveal a representative
and meaningful average picture of the synoptic flow conditions for all interaction categories. These conditions are distinct
between the interaction types yet show similar structures for both ocean domains. Therefore, results are shown here for the
North Pacific domain only (and the North Atlantic results are given in the Supplement). Also, for the sake of conciseness,
the discussion here focuses on the composites for ridge and block interaction types, because the ridge interaction type is the
most frequent one, and the block interaction type shows the most interesting differences. Results for the other interaction types
are also briefly discussed but shown in the Supplement (Fig. S14, S15). Composites will be shown for anomalies of upper-
level PV, mid-tropospheric geopotential height, EKE, and the low-pass (10 days) filtered upper-level zonal winds. Together,
these variables illustrate both the ambient low-frequency background flow and the high-frequency activity associated with the
different interaction types and in the different WCB phases.

## 5.1 Ridge-interaction flow conditions

We first look at the ambient conditions prevalent during ridge interactions (Fig. 9). The upper-level synoptic conditions at the
start-of-ascent feature a Rossby wave pattern with a ridge in the west of the domain and a broader trough downstream, as
indicated by the PV and geopotential height anomalies (Fig. 9a). The WCB air parcels (green contours) start their ascent south
of this ridge and gradually move northward as they ascend. The upper-level pattern propagates slightly eastward as the ascent
progresses to the mid-ascent phase, as expected of upper-level features embedded in westerly flow (Fig. 9b). As the WCB
reaches the end-of-ascent, the ridge anomaly propagates to the central part of the domain, with most WCB air parcels arriving
within the ridge (Fig. 9c). At the point of interaction, most of these WCB air parcels are still within the ridge. Both the WCB
outflow and the ridge are located slightly further east compared to the end-of-ascent (Fig. 9d). At this stage, the meridional PV
anomaly dipole indicates a strengthening of the westerly jet in the interaction region of the WCB.

Throughout this evolution, the upper-level EKE anomaly exhibits high positive values over the ocean domain, indicating
that the synoptic activity over the region is enhanced compared to the DJF climatology. The high EKE anomaly values also
shift slightly eastward as the WCB air parcels progress from start-of-ascent to point-of-interaction (Fig. 9e-h). This increased
synoptic activity could be related to the intensification of the jet via the enhanced PV gradient, which in turn is related to the
low-PV outflow of the WCB air parcels (Fig. 9g,h).

In summary, the ridge interactions are preceded by a non-stationary wave pattern and increased synoptic activity in the
upper troposphere, at least partly related to the increased meridional PV gradient and associated intensification of the zonal





flow. The WCB outflow reaches the upper-level ridge during its end-of-ascent and continues to remain within the ridge during the interaction. In the late WCB phases, the EKE anomaly maximum coincides with the frequency maximum of the WCB air parcels, indicating that upper-tropospheric EKE can be directly influenced by airstreams with intense diabatic processes. Similar patterns for ridge interactions are observed in the North Atlantic domain, where EKE anomalies are strengthened as the WCB outflow reaches the ridge (Fig. S16).

## 5.2 Block-interaction flow conditions

The composites for the block interaction type show some interesting differences. At the start-of-ascent, an intense ridge, indicated by the strongly negative PV anomaly, is already located over the North Pacific between 40-60°N (Fig. 10a). The large positive geopotential height anomaly also reveals this intense upper-level ridge. Troughs are located south and downstream of the ridge, and the WCB air parcels begin their ascent southwest of this prominent structure. The upper-level pattern is fairly stationary during the ascent of the WCB (Fig. 10b), which ends in the upper-level ridge, thereby slightly intensifying its spatial extent and geopotential height anomaly (Fig. 10c). At the point-of-interaction, the intense negative PV anomaly and the negative geopotential height anomaly are maintained and remain collocated with the WCB outflow (Fig. 10a). The geopotential height anomalies at this stage illustrate a diffluent blocking pattern with the trough to the south of the huge ridge anomaly.

During all WCB ascent phases, the dipolar EKE anomalies indicate a poleward shift in synoptic activity, with values exceeding the climatological average over the poleward half of the ridge and lower values along its equatorward side (Fig. 10e-h). This is consistent with where the ridge intensifies and weakens the westerly flow, respectively. Furthermore, due to the presence of the prominent ridge, the eastward extent of the jet streak is restricted, and the (weak) westerlies over the eastern Pacific are shifted equatorward (brown contours, Fig. 10e-h). The WCB air parcels end their ascent at higher latitudes (compared to the ridge-interaction WCBs) where the westerlies are comparatively weak, preventing their downstream advection. The strong anticyclonic circulation associated with the block that remains almost stationary during the period from end-of-ascent to the point-of-interaction, also constrains the WCB air parcels within the block.

In summary, an elevated synoptic activity with large-scale circulation anomalies, pointing to a strongly perturbed waveguide, a pre-existing ridge, and the outflow poleward of the westerly jet are typical for the block-interaction type of WCBs. Over the North Atlantic, comparable flow anomaly patterns were found with the pronounced pre-existing negative PV anomaly, which gets intensified by the WCB outflow (Fig. S17). Additionally, the reduced eastward jet extension is even more evident in the North Atlantic composite (Fig. S17).

## 5.3 No-interaction flow conditions

Even though the no-interaction scenario may seem the least interesting in terms of synoptic dynamics and downstream impact, composites of these events offer valuable insights into the factors that restrain WCBs from producing waveguide disturbances. During the start-of-ascent, the ambient conditions preceding the no-interaction exhibit very weak geopotential and PV anomalies, with a weak trough-ridge pattern over the central North Pacific (Fig. S14a-d). The upper-level pattern remains largely




**Figure 10.** As Fig. 9, but for block interaction time steps.





unchanged as the WCB air parcels ascend to mid-levels. As the end-of-ascent is reached, some air parcels enter the down-stream ridge over the eastern North Pacific, leading to its slight amplification.

The wave pattern amplifies at the time of the interaction point, but the WCB air parcels are advected eastward out of the ridge, some of them rapidly moving along the western flank of the downstream trough over North America. This results in the large eastward and equatorward shift of the no-interaction WCB outflow compared to the other categories discussed in Sect. 3.

As expected of no interaction, the PV anomalies are insignificant within the outflow at the point of interaction.

EKE anomalies are negative during all WCB phases, which implies that the synoptic activity over the domain is lower than in the DJF climatology. In other words, the waveguide is less disturbed, and the synoptic eddies are comparatively weak (Fig. S14e-h). The WCB air parcels end their ascent within the low-pass filtered westerly jet and, therefore, get strongly advected until the point-of-interaction.

In short, the ambient conditions for no-interactions are characterized by a steady waveguide and low synoptic wave activity and the WCB outflow reaching strong westerlies are unfavorable for the WCB to perturb the waveguide. As a consequence, the WCB air parcels are rapidly advected downstream along the only weakly perturbed upper-level flow. Interestingly, the low-frequency jet maximum and the WCB outflow position are almost identical for the no-interaction and ridge categories. This points to the importance of the preceding state of the waveguide, i.e., whether it is already perturbed (as for the ridge-

interaction) or stable (as for the no-interaction), in determining the type of WCB interaction. Similar characteristics of the no-interaction type, such as reduced EKE and WCB outflow within the westerlies, are also found in the North Atlantic composite (Fig. S18).

## 5.4  Cutoff-interaction flow conditions

Similar to block interactions, an intense negative PV anomaly and strong ridge are present at the start-of-ascent of the cutoff-

interaction (Fig. S15a). However, in this case, the ridge is zonally much more confined and flanked by two positive PV anomalies upstream and downstream, respectively. The PV anomaly propagates poleward from the start-of-ascent phase to end-of-ascent. The corresponding ridge gets detached from the wave train at the end-of-ascent, with WCB outflow reaching within the ridge (Fig. S15c). Later, until the point-of-interaction, the negative PV anomaly moves westward, as seen before in the climatology (Fig. 6a,b).

The end-of-ascent region of the cutoff-interaction WCBs is located far poleward of the low-frequency westerly jet (Fig. S15g) This, together with the amplified synoptic-scale wave disturbance in the form of a pre-existing ridge, enables the further poleward motion of the WCB air parcels and the cutoff formation. The elevated synoptic activity is also represented by the positive EKE anomalies in the early phases of the WCB ascent (Fig. S15e,f). In the later stages of the ascent, when the low-PV cutoff forms far poleward of the main jet and thereby reduces the jet intensity via its anticyclonic circulation, the EKE

anomalies weaken and eventually become negative. This is consistent with the relaxation of the undulations in the waveguide and the meridionally oriented PV anomalies at the point-of-interaction.

In summary, WCB cutoff-interactions are favored if the antecedent upper-level flow conditions feature a strongly perturbed waveguide with a high-amplitude ridge, and the WCB outflow reaches into this pre-existing ridge far poleward of the westerlies.



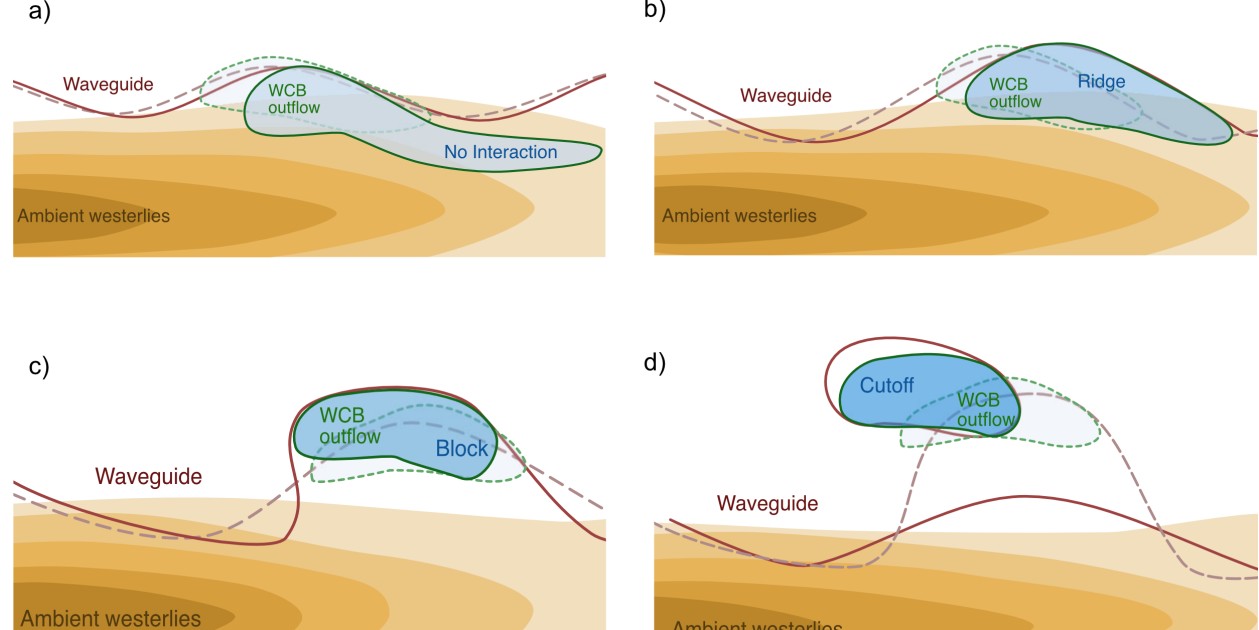

**Figure 11.** Schematics of the four types of WCB-waveguide interactions: **(a)** no interaction, **(b)** ridge, **(c)** block, and **(d)** cutoff interactions. Yellow shading represents the low-frequency upper-level westerlies (darker shading for higher wind speed). Dashed and solid red lines show the instantaneous waveguide at the start-of-ascent and at the point-of-interaction, respectively. Dashed and solid green lines denote the WCB positions at the end-of-ascent and at the point-of-interaction, respectively. Blue shading illustrates the negative upper-level PV anomaly (darker for more negative anomalies).

These characteristics are also observed in the cutoff-interactions over the North Atlantic (Fig. S19). Contrary to the North
Pacific domain, which primarily supports cyclonic wave breaking near the Bering Sea, the North Atlantic domain supports
both cyclonic wave breaking near Greenland or anticyclonic wave breaking over Europe, consistent with Davini et al. (2012).

## 6   Conclusions

This study used ERA5 reanalyses to systematically classify the interaction of WCB outflows with the upper-level Rossby
waveguide. We present an objective approach to classify WCB-waveguide interactions, compile a first global climatology of
these interactions for the period 1980–2022, and explore their characteristics and dynamics. With our approach, interactions
are categorized into four types: (a) no-interaction, (b) ridge-interaction, (c) block-interaction, and (d) (tropospheric) cutoff-
interaction. The no-interaction type also contains events with weak perturbations of the waveguide, which were not strong
enough to be classified as a ridge-interaction. The categories are determined by a combined investigation of the prevailing
upper-level weather features in the WCB outflow region, and of the outflows' PV anomaly values. Based on the intensity of
the PV anomalies and the latitude of the WCB outflows, a hierarchy can be established, ranging from no-interaction to ridge,




block, and, the most intense interaction type, cutoff-interaction. As in previous studies, a Lagrangian approach is employed to identify WCBs, and specifically four stages along their ascent, from start-of-ascent to mid-ascent, end-of-ascent, and eventually point-of-interaction. As outlined in detail in Sect. 2, trajectories are calculated backward for five days from the region poleward of the waveguide to systematically capture all WCB-waveguide interactions. The methods used in this analysis have certain
subjective elements (as explained in Sect. 2), and the sensitivity of the main results to these choices has been tested to be weak.

The main findings of our WCB-waveguide interaction climatology, addressing the research questions introduced in Sect. 1, can be summarized as follows:

(i) What is the relative frequency of the four interaction types, and where do they occur relative to the climatological waveguide?

- The climatological analysis in the NH boreal winter shows that tropospheric WCB outflows most frequently result in ridge interactions (58.7%), followed by no-interaction (27.7%), block (9.7%), and cutoff (3.9%) interactions.

   - These interaction types exhibit a distinct spatial preference, with the interaction regions of the no-interaction type predominantly observed equatorward of the climatological waveguide, the ridge-interactions along and poleward of the waveguide, followed by block, and cutoff interactions, the latter being the most poleward (Fig. 5, 6a). The poleward shift
in the interaction locations is accompanied by a systematic westward displacement, indicative of a reduced influence of the upper-level westerly jet.

   - A similar, but less strong, poleward and westward shift from the no-interaction to the cutoff-interaction type is also observed at the end-of-ascent locations (Fig. 6b). Compared to these locations, the interaction regions for the no-interaction and ridge interactions show a strong eastward advection after the end-of-ascent. In contrast, the block and cutoff interac-
tions have similar interaction and end-of-ascent regions, revealing diminished influence of advection by the westerlies.

   - In contrast, the start-of-ascent and mid-ascent regions are remarkably similar across all interaction categories (Fig. 6c,d). These findings highlight that the large differences in the outflow latitude between the four interaction types are independent of the initial ascent location. This emphasizes the importance of accurately representing outflow position in forecasting, as it directly influences the interaction type and the downstream flow evolution, consistent with the findings
of Martínez-Alvarado et al. (2016), Madonna et al. (2015), and Grams et al. (2018).

(ii) How do the WCB characteristics differ between the interaction types?

   - The average WCB outflow age, i.e., the average time since the end-of-ascent, exhibits a distinctive pattern for the different interaction types (Fig. 7). No-interaction and cutoff interactions are primarily associated with older outflow, while relatively younger outflow leads to ridge and block interactions.

- The WCBs associated with different interaction types exhibit distinct characteristics at the end-of-ascent and the point-of-interaction (Fig. 8). WCB outflows involved in stronger interactions (such as block and cutoff) reach lower pressure



levels and have more negative PV anomalies in both phases. In contrast, the outflows leading to no-interaction are characterized by weaker PV anomalies and higher pressure values. Between the end-of-ascent and the point of interaction, no-interaction air parcels tend to descend and lose PV anomaly strength. In contrast, the outflows leading to intense inter-
action largely maintain the anomaly values between the two phases. It is insightful to reconsider the quantitative values: at the point-of-interaction, negative PV anomalies of ridge, block, and cutoff interactions exceed an amplitude of 2 PVU. In contrast, these anomalies are only about $-0.5$ PVU for what we classify as no-interaction. This large difference in the amplitude of the imposed negative PV anomaly provides a posteriori justification for categorizing these WCBs as non-interacting.

(iii) How do the ambient flow structures differ between the interaction types?

- The composite analysis demonstrates that the synoptic situations for the interaction types differ significantly, indicating the important influence of the large-scale flow on the interaction (Sect. 5). The schematic (Fig. 11) illustrates the key features and differences across interaction types. The no-interaction type is preceded by reduced synoptic activity (negative EKE anomalies), denoting a largely undisturbed waveguide (Fig. 11a, dashed red line). Furthermore, no-interaction
WCBs end their ascent equatorward of the waveguide within the low-frequency westerlies (yellow shading). Conversely, ridge interactions occur when the ambient conditions exhibit positive EKE anomalies, indicating increased synoptic activity with a perturbed waveguide (Fig. 11b). Although the end-of-ascent positions are also within the low-frequency westerlies, the perturbed state of the waveguide and higher PV anomaly values enable the outflow to amplify the ridge (Fig. 11b, solid red line) rather than being advected by the westerlies.

- The block interactions are favored when a pre-existing ridge (or potentially a block itself; Fig. 11c) is present, indicating a highly perturbed waveguide. The outflow positions are poleward of the low-frequency westerlies, and the strong anticyclonic circulation associated with the block prevents the downstream advection of the interacting WCB air parcels. The cutoff interactions are similarly preceded by a prominent ridge and high synoptic activity (Fig. 11d). The outflow associated with cutoff interactions is positioned further poleward, with minimal influence from the westerlies. As the
tropospheric cutoff forms, the waveguide relaxes, leading to lower EKE values over that region.

(iv) Is there seasonal and/or hemispheric variability in WCB interaction types?

- The above results are consistent in all seasons and hemispheres (as documented in detail in the Supplement and briefly discussed in Sect. 3.3). In the NH, the frequency of the interactions varies with seasonal changes in WCB activity, but the relative frequencies remain similar, with ridge interactions being the most common and cutoff interactions the least
common. The primary interaction regions for the different interaction types also vary slightly with season, owing to the seasonal changes in the occurrence of the associated weather features and the strength of the westerlies. In the SH, the interaction regions are extensive, almost covering the entire mid-latitude band.

In summary, we found that the interaction types of WCB outflows with upper-level waveguides differ strongly in terms of the outflow position relative to the waveguide and the ambient upper-level flow conditions. In contrast, the inflow regions and



the properties of the WCBs prior to the ascent are fairly similar for all interaction types. Weak or no interactions typically occur when the upper-level waveguide is relatively zonal with low synoptic activity, and the WCB outflow remains equatorward of the ambient westerlies. In contrast, intense interactions, such as low-PV cutoff formations or block amplifications, arise when the upper-level flow exhibits high synoptic activity with a disturbed waveguide already at the start-of-ascent of the WCBs, together with poleward WCB outflows that feature intense negative PV anomalies. A limitation of our study is the subjectivity involved in some of the methodological choices. They include the duration of the backward trajectories, the definition of ridges,

and the thresholds used in composite analysis. However, we tested for the sensitivity to meaningful variations of these choices and these tests confirmed the robustness of the main results summarized above.

  In the final paragraphs, we would like to discuss our results in the broader context and mention some suggestions for further research. Our study emphasizes the significant influence of ambient synoptic conditions on the interaction between WCB

outflows and the Rossby waveguide. Nonetheless, this does not lessen the significance of diabatic processes, as this study a priori focuses only on periods of strong diabatic activity when WCBs are present. Specifically, the cross-isentropic ascent of WCBs and the resulting PV anomalies in the WCB outflow are heavily influenced by microphysical processes (Joos and Wernli, 2012). Further, latent heating is essential for enabling divergent outflows to reach upper levels with low PV values. As noted by Steinfeld et al. (2020), diabatic processes provide necessary amplification of anticyclonic flow anomalies alongside

dry-dynamical forcing, which is consistent with our findings.

  Details of the microphysical processes can also influence the outflow region (Joos and Forbes, 2016), and diabatic processes have been shown to also influence the cyclone propagation direction (Tamarin and Kaspi, 2016), which in turn affects the ascent of the subsequent WCB trajectories. The latter study showed that cyclones with significant latent heat release tend to move more poleward. Investigating in future studies the cyclone properties associated with each WCB-waveguide interaction type,

such as their propagation direction, could further elucidate whether cyclone properties differ systematically between interaction types.

  The study also opens avenues for several important future research directions regarding the evolution of WCB-waveguide interactions and their predictability. The accurate representation of the WCB-waveguide interactions in numerical models is most likely essential, as systematic forecast errors often occur in the presence of such intense diabatic outflows (Rodwell et al.,

2013; Gray et al., 2014). Pickl et al. (2023) found that forecast skill generally decreases when WCB activity is high, and WCB activity significantly increases when error growth is largest. In the two case studies of forecast bust over the Atlantic-European domain analyzed by Martínez-Alvarado et al. (2016) and Grams et al. (2018), the forecast errors were attributed to the inaccurate position of WCB outflow, which was predicted too far south, and resulted in an underestimation of the PV anomaly. Forecast errors caused by diabatic outflow can amplify and propagate downstream along the waveguide, eventually

affecting predictability in the downstream region (Rodwell et al., 2013; Grams et al., 2018; Pickl et al., 2023). Thus, it would be rewarding to quantify forecast uncertainties and errors specifically during the four WCB-waveguide interaction types.

  Further research could also explore the influence of synoptic conditions in idealized simulations, to clarify the complex interplay between WCB outflows and the waveguide, especially given that the waveguide is often already disturbed in the real world at the start-of-ascent of WCBs. Such simulations could systematically investigate the evolution of WCB-induced upper-



level flow anomalies for different jet strengths and jet-anomaly configurations. These new research directions and questions could further advance the theoretical understanding of WCB-waveguide interactions, with an ultimate goal to improve the predictability of these interactions, the downstream flow evolution, and associated weather events.

*Data availability.* The ERA5 datasets are available from the Copernicus Climate Change Service (C3S) Climate Data Store at https://doi.org/10.24381/cds.143582cf (Hersbach et al., 2017). Other data from this study can be obtained from the authors upon request.

*Author contributions.* VS performed the analyses and wrote the first version of this paper. MS, HJ, and HW helped VS in the design of the study, the interpretation of the results, and the writing of the paper.

*Competing interests.* At least one of the (co-)authors is a member of the editorial board of Weather and Climate Dynamics. The authors also have no other competing interests to declare.

*Financial support.* This research has been supported by the Schweizerischer Nationalfonds zur Förderung der Wissenschaftlichen Forschung (grant no. 185049).

*Acknowledgements.* We thank the members of the Atmospheric Dynamics group at ETH Zurich, especially Franco Lee, for their insightful discussions and Nora Zilibotti for providing the EKE data.



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
