# Peer review of "The interaction of warm conveyor belt outflows with the upper-level waveguide: a four-type climatological classification"

_EGUsphere, 2025_

## Referee Comment (RC2)

**Review of *"The interaction of warm conveyor belt outflows with the upper-level waveguide: a four-type climatological classification"* by Vishnupriya et al.**

**Summary**

Vishnupriya et al. present a diagnostic study that classifies warm conveyor belt outflow interactions with the upper-level jet stream (Rossby waveguide). Using ERA5 and Lagrangian tracking, the authors systematically combine multi-decade seasonal climatology of WCB–waveguide interaction types, extending prior case-specific research by introducing an objective classification. The large sample lends confidence that the reported frequencies and patterns are robust climate statistics rather than anecdotal findings. The methodology, from using the well-known LAGRANTO tool for WCB tracking, to identification of waveguide disturbances, is rigorous and consistent with previous literature. The case studies (Fig. 4) and schematic drawing (Fig. 11) are helpful in argumentative demonstrations. Despite the complexity of the subject, the manuscript is generally well-organized and written. Overall, the paper's structure (methods, case examples, climatology, composites, summary) makes it easy to follow the logical progression from methodology to key conclusions. Therefore it has the potential to be published in WCD, but some clarifications and revisions are still needed.

**Major Comments**

- **Threshold Sensitivity.** The classification relies on specific PV threshold criteria that, while grounded in prior studies, are somewhat subjective. For example, ridges are defined by a PV anomaly < –1 PVU and cutoffs by PV < 2 PVU. Likewise, the blocking definition requires a –1.3 PVU anomaly persisting 5 days. It is not fully explored how sensitive the results (especially the relative frequencies of interaction types) are to these threshold values. The authors note that certain methodological elements are subjective but claim the main results are not sensitive to those choices; however, the paper would benefit from evidence of this (e.g. a brief sensitivity test varying the PV anomaly cutoff by some amount). As it stands, it is hard to know if slightly different thresholds might change an event from "weak interaction" to "ridge" or alter the 58.7% ridge frequency.

- **What about the stratosphere?** In the conclusion part (line 585-586), the authors find that in boreal winter, "tropospheric WCB outflows most frequently result in ridge interactions (58.7%), followed by no-interaction (27.7%), block (9.7%), and cutoff (3.9%)

interactions." However, such numbers come from a normalization stated in Section 3 (line 250-252): "ridge interacting type (54.0%), followed by no interaction (25.5%), while block (8.9%) and cutoff (3.6%)". So there are about 100% - (54 + 25.5 + 8.9 + 3.6)% = 8% of stratospheric interactions resulting from WCB outflows that are excluded from this study. This 8% is comparable to the 8.9% blocking type and way larger than the 3.6% cutoff type. While it's understandable to focus on tropospheric impacts, these excluded cases (gray dots in their Fig. 2b) are interesting for completeness, especially given that some literatures argue stratosphere-troposphere coupling may be important and modifying blocking frequency (e.g., Davini et al., 2014), which potentially indicates the left-out stratospheric outflows to be a non-negligible category. Hence, it'd be best for the authors to answer if stratospheric WCB outflows are truly negligible in number and in analyses? Also it'd be best for the authors to include the "tropospheric" constraint in line 15 for WCB outflows.

- **A block could be double counted as a ridge!** Computationally, the authors still enforce mutual exclusiveness by applying a strict hierarchy when they tag each trajectory, so every WCB parcel ends up in one and only one class. Yet, a single trajectory's starting point can coincide with multiple feature types, say a block would typically also satisfy a ridge criteria. This hierarchy, while physically reasoned (a cutoff indicates a more intense wave breaking than a ridge), could lead to ambiguous cases being classified as the highest-ranking feature present. It would be useful to clarify whether the hierarchy ever overrides what a meteorologist might consider the primary interaction. The authors should ensure this automated decision-making doesn't misclassify borderline situations or transitions of types, such as an initial ridge may become a block *after* temporal persistence threshold being met. What's the proportion of such borderline transitions in the study? Wouldn't this also lead to a developed block being slightly older than spawned ridges, in contrast to Fig. 7 panel c, where block is argued to be of the youngest WCB outflow age at the point of interaction? For air parcels in a block, how many of them are of "young age" captured in Fig. 7c, how many of them have stayed in the upper level for more than 5 days (not captured in Fig. 7c)?

- **Timing of Interaction Assessment.** The classification is determined at the backward trajectory starting point, which is termed the "point of interaction," but this choice may not capture delayed or downstream impacts. If a WCB outflow does not immediately coincide with a ridge/block at the initial time but contributes to one a day later, such an effect would be missed by the algorithm. In other words, some WCBs might be labeled "no interaction" at the start point but go on to amplify a wave downstream. The paper would benefit from a discussion of this limitation – essentially, the method diagnoses interactions at a fixed time (when trajectories hit the 2-PVU surface) and may not track the subsequent evolution. This is partly addressed by analyzing the eastward advection of no-interaction vs. stagnation of block-type air parcels, but it remains possible that a WCB initially categorized as non-interacting could induce a ridge slightly later. Clarifying

how the results might change if the interaction were evaluated over a time window (rather than an instantaneous point) would be useful.

- **Causality vs Colocation:** The study assumes that if a WCB outflow is co-located with a ridge, block, or cutoff, then the WCB *interacts* with the waveguide to produce that feature. This is a reasonable interpretation, but it is essentially inferred rather than directly proven. The authors offer a posteriori justification by noting the much weaker PV anomalies for trajectories classified as non-interacting, implying those indeed had minimal effect. Still, the methodology identifies *associations* between WCB outflows and PV disturbances – it does not demonstrate that the WCB caused the disturbance. It would strengthen the paper if the authors could argue more explicitly that the identified ridges/blocks were actually enhanced by the WCB. As an example, one could ask: might a ridge have existed anyway, with or without the WCB, and the WCB simply happened under it? The implicit assumption of causality could be more critically discussed. This point is important for interpreting the climatology: the paper shows where WCBs *co-occur* with certain waveguide disruptions, but future work (perhaps using modeling experiments) would be needed to confirm the extent of the WCB's causal influence.

- **Statistical / significance testing.** While the dataset is large, the manuscript does not report formal significance testing for differences between categories. Phrases like "differ significantly" (line 616) in the composite analysis appear to be used qualitatively. It would improve the rigor if the authors could demonstrate that key distinctions (e.g. the differences in outflow latitude or PV between categories) are statistically significant. Similarly, for the composite maps (Fig. 9 and 10), showing stippling or some indication of significance for anomalies would help support statements that, say, blocks are preceded by significantly higher eddy kinetic energy than no-interaction cases. If significance testing was not conducted, the authors should temper the language or clarify that "significantly" is meant in a qualitative sense. Adding some basic statistical analysis would bolster the conclusions that the observed differences are robust and not artifacts of variability.

- **What happens from Mid Ascent to End of Ascent?** One thing stands out in Fig. 6 is that the four types of interactions could hardly be distinguished from each other in panel 6c Mid Ascent, but there are significant spatial differences demonstrated in panel 6b End of Ascent. What in this ascending process is causing the difference? Could you connect Fig. 9 and Fig. 10 arguments with Fig. 6 panel b and c differences? If possible, could you expand in details about the governing mechanism?

- **Composite Selection Bias:** The method for constructing composites introduces additional subjective criteria: the authors only composite time steps when a given interaction type is sufficiently dominant (>40% of WCB trajectories in the region). This ensures "pure" cases but might bias the composites towards extreme examples. For instance, a time step with 39% no-interaction, 46% ridge, 10% block, 5% cutoff might be excluded entirely, whereas a time step with 41% no-interaction triggers inclusion as a "no-interaction case". Such hard thresholds (40% for one type, and a secondary 10% cutoff criterion for the cutoff type) could skew the sample of events used for composites. The authors should justify the choice of 40% – presumably to get a decent sample size while maintaining category signal – and perhaps test that varying this threshold (30% vs 50%) does not qualitatively change the composite patterns. If a more inclusive compositing approach yields similar patterns, that would alleviate concern that the composite results are dependent on this filtering.

**Minor Comments**

- **Fig. 4.** The abscissa and ordinate in Fig. 4 are wrong. You cannot have two 80°N in one map, nor 0-60°W being perpendicular to 0-80°E.

- **Fig. 8.** Panel a does not explain the letter M in the total number of trajectories.

- **Fig. 9-10.** The caption of labelling (a,e) as point of interaction is not consistent with the figure labelling (a,e) as start of ascent. Same inconsistency happens for all rows in the plots.

- **Composites of cutoff-interaction time steps should be included in the manuscript, not in supplementary materials**. The difference for block interactions to ridge of having "intense negative PV anomaly and strong ridge" (line 549) are important, but the cutoff interactions also share this difference and signify a more important sign change of EKE anomaly throughout the ascending time range. I would suggest swap Fig. S15 with Fig. 10.

**Recommendation:** *Major Revision*.

---

## Author Comment (AC1)

**Final author comments for egusphere-2025-1731**

**The interaction of warm conveyor belt outflows with the upper-level waveguide: a four-type climatological classification**

by Selvakumar Vishnupriya, Michael Sprenger, Hanna Joos, and Heini Wernli

03 July 2025

We are most grateful to both reviewers for their detailed and constructive comments that help us to further improve the manuscript. Based on the reviewers' suggestions, we will implement several changes in the manuscript. The main changes are that:

- We better explain our terminology ("interaction") to avoid misunderstandings about causality. In line with the view of the reviewers, we regard warm conveyor belts (WCBs) as inherent elements of moist baroclinic waves.
- We acknowledge that, with our approach, we cannot distinguish between WCB-waveguide interactions with a preexisting ridge and those where the ridge only forms during the interaction.
- We perform statistical analyses on the composites in Sect. 5 to assess the robustness of the differences, particularly for PV and EKE found in the four categories of WCB-waveguide interactions.
- We also perform a sensitivity study regarding the thresholds chosen to sort individual timesteps into the four categories of WCB-waveguide interactions.

This document presents the reviewers' comments in **blue** and our responses in **black**.

**Reviewer 1**
The manuscript by Vishnupriya et al. considers an important topic: the interaction of moist-baroclinic development with the evolving larger-scale midlatitude circulation. The authors phrase this topic in terms of the interaction of warm-conveyor-belt (WCB) outflow with the midaltitude waveguide and examine the climatological behavior over a 43-year period using ERA5 re-analysis data. Four important classes of interaction are (subjectively) classified and WCB outflow and antecedent flow conditions examined. The authors find differences in WCB outflow characteristics that are consistent with the subsequent evolution of the larger-scale flow and document the important insight that the flow evolution following WCB evolution is largely dependent by the preceding large-scale conditions.

This study fits very well in the scope of the journal and improves our understanding of the sensitivity of the midlatitude, larger-scale flow to WCB interactions, which may have important implications for predictability aspects, as noted by the authors. Overall, the manuscript is well written with informative figures. I am critical, however, about a central concept of the study: the definition of the WCB-waveguide interaction, which has implications for causality statements, and which is left implicit in the manuscript. Related, the presentation of WCB as atmospheric features leaves room for the interpretation that WCBs and their outflow have 'a life of their own' and can be

considered as 'external perturbations' to the midlatitude circulation, whereas in fact they are intrinsically tied to moist-baroclinic development embedded within the coupled eddy-driven jet – synoptic eddies system that is the midlatitude storm tracks. While the latter point may be a matter of style and perspective, I think that the manuscript will benefit from de-emphasizing WCBs as independent features and emphasizing the coupling of processes in the storm tracks.

I recommend major revisions before publication.

Best wishes!

We sincerely thank Reviewer 1 for their thoughtful and constructive comments, as well as their positive assessment of the manuscript's significance and clarity. We greatly appreciate the time and effort dedicated to reviewing our work and the insightful suggestions provided. We hope the revised version will match your expectations. Below, we provide detailed, point-by-point responses to each comment, outlining how we plan to revise the manuscript accordingly.

General comments:
- **Concept of WCB-waveguide interaction**

Trajectories, by their very definition, follow the ambient flow. They are invaluable in identifying coherent air streams and processes within these moving air masses. In isolation, however, trajectories do not provide information about *why* the ambient flow evolves as it does, i.e., trajectories provide in this respect limited information about causality.
The term "WCB-waveguide interaction" strongly implies causality: The WCB "acts" on the waveguide (and vice versa). Throughout their manuscript, the authors illustrate that the WCB outflow after the end of ascent follows the upper-tropospheric flow: If there is a ridge, the outflow fills the ridge and "older" outflow air is advected further downstream; if there is a cut-off, the outflow air is trapped in that cut-off, … The authors' schematic Fig. 11 makes this notion quite explicit. Do the authors consider this advection as the action of the waveguide on the WCB, i.e., as part of the interaction? Or what is the action of the waveguide on the WCB?

We thank the reviewer for this important point. We consider the WCB–waveguide relationship as a two-way interaction, which does not necessarily imply causality. Our classification captures the combined WCB-waveguide flow patterns and the resulting waveguide disturbance, rather than providing a direct, isolated measure of forcing by the WCB alone. WCB outflows play a crucial role in generating negative potential vorticity (PV) anomalies, which are then rearranged by the ambient upper-level flow. This dynamical interplay shapes different interaction types. Thus, our approach captures both the immediate interaction, where WCB outflows directly influence the waveguide, and the delayed impact, when these PV anomalies evolve within and modify the larger-scale flow.
In this sense, the term "interaction" conveys the *mutual influence* between WCB outflow and the ambient waveguide flow, acknowledging that the waveguide also acts upon and advects the WCB outflow. We will clarify in the manuscript that the term "interaction" as used here does not imply strict one-directional causality. Rather, it denotes the co-occurrence and dynamical linkage between WCB outflows and large-scale waveguide features, without attributing direct cause-effect relationships.

The authors define "point-of-interaction" as the start of the backward trajectories, which may be up to 3 days after the end of the ascent, i.e., may have traveled rather passively for up to 3 days. What is the nature of the action of the WCB outflow on the waveguide at this point? I might be wrong,

but my answer is: There is no action, except possible due to a modification of the radiative properties of relatively moist and cloudy "young" outflow air. Or do the authors have in mind the (usually small) difference of PV values of "young" outflow air and the ambient low-PV air *equatorward* of the waveguide?

We thank the reviewer for raising this question. We acknowledge that the WCB outflow air parcels may indeed be passively advected by the upper-level flow, and the direct diabatic forcing associated with ascent may have ceased at the point of interaction. However, note also that the WCB outflow typically leads to intense negative PV anomalies (see, e.g., Sect. 5.5 in Wernli and Gray, 2024); clearly, these anomalies interact with and influence the evolution of the PV waveguide. Therefore, there can be direct "action" (to use the terminology of the reviewer) of the WCB outflow on the waveguide. Our approach focuses on the evolution of PV anomalies generated by the WCB outflow, which continue to influence the waveguide well beyond the end of ascent. The "point-of-interaction," therefore, represents a stage when these PV anomalies interact with and modify the larger-scale flow pattern, contributing to different flow configurations such as ridges, blocks, and cutoffs. While the immediate diabatic forcing is strongest near the end of ascent, the interactions are also driven by the dynamics of the WCB-induced negative PV anomalies with the ambient flow during the days after the ascent.
We do not attribute the interaction at this stage to radiative effects of moist or cloudy outflow air, but rather to PV anomalies associated with WCB end-of-ascent that persist and evolve within the waveguide.

Much previous work, including work in the authors' group, have argued that a strong action on the waveguide occurs where the outflow is (actually) horizontally divergent. Archambault et al. (2013) explicitly defined an interaction metric based on PV advection by the divergent wind, the divergent wind provides forcing terms in PV budgets of upper-tropospheric PV anomalies (e.g., Teubler and Riemer 2021), and the authors' group has in previous work indicated on maps the locations where WCB trajectories cross upper-tropospheric isentropic surfaces (cross-isentropic transport relates to horizontal divergence by continuity and approximately vanishing vertical motion at the tropopause). By continuity, horizontal divergence $\partial u/\partial x + \partial v/\partial y = - \partial \text{omega}/\partial p$. From the authors' schematic Fig. 3, horizontal divergence is maximized near the end of the ascent, whereas horizontal divergence vanishes for the point of interaction. My specific suggestion is to use the end of ascent as point of interaction, which is physically more justified and should exhibit little sensitivity to reasonable choices of the length of the backward trajectories. In the current manuscript, analyzing the time between end of ascent and "point of interaction" (e.g. in Sect. 4) merely serves to sample the emergent flow pattern without providing a causal link from WCB to flow pattern.In fact, at the end of section 3.2 the authors make a very helpful statement: "This will help us better understand how the properties of the WCB air parcels at the end-of-ascent and the ambient flow conditions together determine the interaction of the WCB outflow with the waveguide." I recommend that the authors frame the purpose of the study more clearly in this sense already in the introduction.
Archambault, H. M., Bosart, L. F., Keyser, D., & Cordeira, J. M. (2013). A climatological analysis of the extratropical flow response to recurring western North Pacific tropical cyclones. *Monthly Weather Review*, *141*(7), 2325-2346.
Teubler, F., & Riemer, M. (2021). Potential-vorticity dynamics of troughs and ridges within Rossby wave packets during a 40-year reanalysis period. *Weather and Climate Dynamics*, *2*(3), 535-559.

We thank the reviewer for this detailed and important comment. We fully agree that the strongest direct forcing of the waveguide by the WCB outflow occurs near the end of ascent, where horizontal wind divergence and associated diabatic cross-isentropic transport are maximized.

In our study, the "point-of-interaction" was chosen to capture not only the immediate outflow forcing near the end of ascent but also the subsequent evolution and rearrangement of PV anomalies within the waveguide flow. While the end-of-ascent marks the peak of the direct forcing, the effects of the WCB outflow on the larger-scale flow can continue to evolve. By including this later stage in our analysis, we aim to characterize the full scope of WCB–waveguide interactions rather than just the initial forcing.

We appreciate the reviewer's suggestion to clarify this framing more explicitly in the Introduction, and we will revise the manuscript accordingly to emphasize our goal and rationale.

On a related note, the use of the term 'interaction intensity' is misleading. While I agree that it is sensible to attribute an 'intensity' to the evolving flow patterns – as in 'strength of the deviation from zonal flow' – the authors have no metric to assess the action of the WCB outflow on the waveguide (in contrast to Archambault et al.). I suggest revising the terminology to avoid confusion. Similarly, I am not sure that the term "interaction types" is helpful terminology. Certainly, WCB outflow occurs and follows different types of flow patterns, but in what sense this represents different types of *interaction* is unclear to me.

We really appreciate the reviewer's perspective on this. We acknowledge that the term "interaction intensity" might suggest a quantifiable measure of the direct forcing of the waveguide by the WCB outflow, which our study does not explicitly provide. The term "interaction intensity" as used in our study reflects a physical progression of ridge-amplification and potentially wave-breaking intensity and was implemented to allow for a consistent categorization of each WCB trajectory and ensure mutual exclusivity of the types. We will explicitly clarify this point in the manuscript to avoid confusion about our terminology.

Similarly, the term "interaction types" is meant to represent distinct flow patterns within the waveguide that are influenced by the WCB outflow. Our intention is to capture the diversity of combined WCB–waveguide evolution scenarios that are associated with different negative PV anomaly features.

- **Implication of causality**
In some parts of the manuscript, the authors imply that differences in WCB outflow are causally linked to the representation of the WCB (e.g., in Sect. 6 around lines 599 and 668, also adopting arguments of previous work). As noted above, trajectories follow the ambient flow and causality cannot be inferred. The WCB will be misrepresented if the ambient flow is misrepresented. A recent study by Oertel et al. found that the impact on the larger-scale downstream flow is dominated by the sensitivity of WCBs to ambient conditions rather than to the representation of the microphysics, consistent with the relatively small impact found by Joos and Forbes (2016).
Please clarify and revise statements implying causality throughout the manuscript.
Oertel, A., Miltenberger, A. K., Grams, C. M., & Hoose, C. (2025). Sensitivities of warm conveyor belt ascent, associated precipitation characteristics and large-scale flow pattern: Insights from a perturbed parameter ensemble. *Quarterly Journal of the Royal Meteorological Society*, e4986.

We appreciate the reviewer's remark regarding the interpretation of causality in WCB–waveguide interactions. We agree that WCB trajectories follow the ambient flow and that causality cannot be strictly inferred from trajectory analysis alone. Our intention was to convey that, since WCB outflow locations differ among the interaction categories, a misrepresentation of the WCB outflow location could lead to a mischaracterization of the type or timing of interaction, ultimately contributing to

forecast errors, as highlighted by studies such as Madonna et al. (2015) and Grams et al. (2018). We also agree with the reviewer that a misrepresentation of the WCB is often a result of a misrepresented upper-level flow. This perspective reinforces our finding that the evolution of WCB outflows is strongly influenced by the surrounding upper-level flow. In the revised version of our manuscript, we will carefully check and amend potentially misleading statements about causality. (A brief remark about the study by Joos and Forbes (2016): in our view, their differences in the WCB outflow due to changes in the model's microphysics are not small, but rather substantial.)

- **WCBs as an intrinsic part of midlatitude dynamics**
WCBs are an intrinsic part of moist-baroclinic growth in the midlatitudes. A few more specific comments relate to this perspective:
i) From this perspective, "WCBs occur all the time" in the midlatitudes and are not "special events" to which the flow would response in specific ways. The main result of the authors, that the impact of WCB interaction depends mostly on the state of the waveguide and to much lesser extent on WCB characteristics, thereby seems very plausible, yet I fully agree that it is worth documenting and supporting by data. In fact, I recommend extending section 5, in which this main result is presented. To me, section 4 mostly illustrated that WCB trajectories *after* ascent merely sample the upper-level flow conditions (as noted above). I thus believe that this section can be streamlined without much loss at the expense of an extended section 5.

Thank you for this insightful and encouraging comment. We appreciate the reviewer's recognition of our key result. We fully agree with the perspective that WCBs are not isolated or exceptional events, but rather a frequent and intrinsic part of moist-baroclinic development in the midlatitudes. However, this study specifically focuses only on the time steps when WCBs occur, and therefore does not address dynamics in periods or regions where WCB outflows are absent.
We also acknowledge that WCB outflow characteristics—such as PV anomalies, outflow latitude, and pressure levels—play a role in shaping the interaction and are themselves influenced by the surrounding synoptic environment. Hence, the interaction outcome reflects a combination of the influence of the waveguide and the evolving properties of the WCB outflow. We will shorten Sect. 4.2 so that readers can focus more on the results in Sect. 5.

ii) Figure 5: My impression of this figure is that we get most of the signal by multiplying the occurrence frequency of WCBs (Fig. 1d) by occurrence frequency of the respective flow pattern (Fig. 1a-c), i.e., simply by combining the occurrence frequencies of two statistically independent events. This impression seems to be supported by the authors description in 3.1. The interpretation is then that e.g., blocks occur with a certain frequency and ridges occur with a certain frequency, but that WCB occurrence is not a discriminating factor between ridges and blocks, which seems to be in some contrast to statements in the introduction that WCBs play an important role in the evolution of certain events. Can the authors comment and clarify?

We thank the reviewer for raising this perspective. We agree that the patterns in Fig. 5 can, to a large extent, be interpreted as a combination of the occurrence frequencies of WCBs and the frequencies of the respective upper-level flow features (as shown in Fig. 1). This is expected, as Fig. 5 illustrates the co-occurrence of WCBs with different flow regimes, i.e., how frequently WCB outflows are associated with each interaction type. It reflects the conditional frequency of WCBs given the presence of certain waveguide features but does not imply causality or exclusivity.
To address the reviewer's concern more directly, we have prepared an additional figure (Fig. R1) comparing the total occurrence frequency of each flow feature with and without WCB involvement. We classify features as WCB-interacting if they contain more than 20 WCB air parcels or if at least

25% of their grid points consist of WCB air parcels. This analysis reveals that the majority of the ridges and blocks interact with WCBs, supporting the conclusion that the low-PV outflow of WCBs could have a significant influence on the evolution of these features. While this analysis is relevant, it is not included in the manuscript because our primary focus is on the *WCBs*, rather than the *features*.

[Figure]

**Fig. R1:** Climatological frequency of flow features during DJF: **(a, d, g)** ridges (at 315 K), **(b, e, h)** blocks, **(c, f, i)** tropospheric cutoffs (at 315 K). (a–c) show the total frequency of these features, (d–f) show the frequency of features interacting with WCBs, and (g–i) show the frequency of non-interacting features.

iii) PV anomaly associated with WCB outflow: "Young" WCB outflow may have different moist/cloud characteristics as ambient upper-tropospheric air masses *equatorward* of the waveguide and may

have somewhat smaller PV values (tenths of PVU). The displacement of the strong PV gradient associated with the waveguide creates very large PV anomalies (several PVU and thus an order of magnitude larger). WCB outflow may flow passively into the region of a large PV anomaly (a ridge) or may actively generate the anomaly by contributing to the displacement of the sharp gradient. Please clarify in the presentation to which type of PV anomaly you refer to and how "passive advection" and "active generation" can be distinguished.

Thank you for raising this important point regarding the nature of the PV anomalies associated with WCB outflows.

The PV anomalies we consider represent how much the PV values of WCB air parcels deviate from the 15-day running mean, both at the point of interaction and at the end of ascent (Sect. 4.2). While it is true that WCB outflows may subsequently be passively advected into regions with strong ambient PV gradients, such as ridges, our analysis—based on the PV characteristics at the end-of-ascent—indicates that the outflow itself is already associated with significant PV anomalies. These anomalies actively contribute to modifying the local PV distribution and, consequently, the Rossby waveguide evolution. We aim to highlight that WCB outflows are typically associated with negative PV anomalies (see also Fig. 8 in Madonna et al., 2014), which can subsequently be advected and contribute to the development or modification of various upper-level flow features.

The additional specific question about distinguishing "passive advection" and "active generation" is interesting and at the same time challenging. We don't think that with our approach we could provide a high-quality answer to this question. Most likely, one would need to quantify PV advection by the irrotational wind and track the PV anomaly features in time. Such an analysis is beyond the scope of our study and must be left for future studies.

- **Concept of age of outflow**
  The significance of this concept not become clear to me.

We appreciate the opportunity to clarify the significance of the *outflow age* concept. In our study, outflow age refers to the time elapsed between the end of WCB ascent and the moment when the WCB air interacts with the upper-level waveguide. This metric is important because it captures the temporal evolution of WCB–waveguide interactions.

Our results show that younger outflows are more commonly associated with ridge and block interactions, while older outflows tend to correspond to cutoff or no-interaction cases. This pattern suggests a typical progression of interaction types over time, from initial ridge or block interaction toward later-stage cutoff or non-interaction, highlighting the dynamic nature of these processes. Additionally, many interaction points include a mixture of younger and older outflows, indicating that different stages of WCB evolution can contribute simultaneously. Thus, the concept of outflow age offers valuable insight into the lifecycle of WCB outflows and their evolving influence on the large-scale flow, including the role of advection by westerlies. We will better explain the motivation for considering this concept in the revised manuscript.

Specific comments:
- L20 westward of what? within individual basins? Not clear at this point.
  Thank you for the comment. In this context, "westward" refers to the relative shift of WCB outflow locations for the different interaction categories. While we acknowledge the ambiguity, we believe the meaning becomes clearer in the broader context of the manuscript. Therefore, we prefer to retain the sentence as it stands.

- L21, "The preceding ..": Relating to one of my general comments: This sentence is very important. The previous presentation may otherwise be misunderstood as WCBs being an external actor on the waveguide and not a feature that develops within the synoptic evolution along the waveguide. The latter perspective could still be made more clearly to further improve the manuscript.

We thank the reviewer for this insightful comment. We agree that it is important to clarify that WCBs are not external actors imposed on the waveguide but rather features that develop within the evolving synoptic-scale flow. The preceding and prevailing synoptic conditions strongly influence how WCBs interact with the upper-level waveguide. We will revise the manuscript to emphasize this perspective more clearly by now writing: "The preceding and prevailing ambient large-scale flow conditions also significantly differ between the interaction types, indicating the large influence of the preexisting synoptic flow situation on how WCBs interact with the upper-level waveguide."

- L32 "the dynamics of upper-level extratropical flows is mostly adiabatic and has comparatively high predictability": The first statement is debatable. *Moist*-baroclinic as the underlying paradigm of the midlatitude circulation dates back at least to the 1970-80's (e.g., Gall 1976, Emanuel et al. 1987). More recently, Teubler and Riemer (2021) used the term *moist*-baroclinic downstream development to emphasize the first-order effect of moist processes. The second statement raises the question: Compared to what? Please clarify this sentence.

We thank the reviewer for this important comment. We fully agree that moist-baroclinic processes are fundamental in midlatitude dynamics, as highlighted by previous studies and the review article by one of the authors. Our original statement aimed to emphasize that the dry, adiabatic component of the upper-level extratropical flow generally exhibits higher predictability compared to the embedded smaller-scale moist processes, thereby emphasizing the importance of further studying these moist dynamical processes. We will clarify this point in the manuscript by specifying the comparison and acknowledging the key role of moist-baroclinic dynamics.

- L80ff: I have no doubt about the usefulness of the authors subjective choice. Just out of curiosity (other readers may be curious, too): Did the authors also think about an objective classification, e.g., based on EOF and cluster analysis?

We acknowledge the reviewer's curiosity. While we recognize the value of objective methods such as EOF or cluster analysis, we did not adopt them in this study. Such techniques typically lead to coarser flow regime classifications and may not reliably capture specific features at the synoptic scale, such as blocks or cutoffs, which are central to our analysis. For our purposes, the feature-based approach provided more direct and interpretable categorization of WCB-waveguide interactions.

- L139: I do not understand, please clarify; the above flow features are not identified in a Lagrangian sense.

Thank you for the comment. We will clarify that the Lagrangian analysis is applied to air parcels in the upper troposphere that are part of the larger Eulerian flow features (e.g., blocks, ridges). The features themselves are not identified in a Lagrangian sense, but WCB trajectories passing through them are tracked using LAGRANTO.

Our revised text reads: "The Lagrangian analysis of air parcels in the upper troposphere, which are associated with the different flow features, can provide comprehensive information about their origin and evolution."

- L152: when --> where

Corrected as suggested — "when" is replaced with "where."

- 319: reveal --> confirm
  Corrected as suggested —"reveal" to "confirm"

- L389: I do not follow this argument. Can you clarify? What is meant with „profit"?
  We appreciate this request for clarification. The intended meaning is that block and ridge interactions occur shortly after the WCB reaches upper levels (as measured by outflow age), suggesting that these interactions are closely linked to the dynamically active stage of the WCB, when its PV anomalies are intense. We will clarify this in the revised manuscript as follows: "We can also interpret the results shown in Fig. 7 in the following way: since young WCB-waveguide interactions occur preferentially in ridges and blocks, the PV anomalies of these flow features are enhanced by the direct injection of low-PV air by WCBs."

- Table 1 and subsection 42.: are described differences in stat sig?
  Regarding Table 1, we do not find significant differences between the values (because the standard deviations are typically larger than the differences between the means), and the purpose of the table is to provide a general comparison of WCB ascent characteristics. For Sect. 4.2, some of the differences, particularly between the no-interaction and cutoff cases, are significant, as visually evident from the boxplots in Fig. 8.

- L410: shift of what?
  Clarified as "shift in the climatological regions of occurrence."

- L426: Why is this "interestingly"?
  "Interestingly" was meant to highlight that block-interacting WCBs show less spread in PV anomaly values. We've revised the sentence to clearly state this observed contrast.
  The revised text reads: "At both considered times, PV values and their variability are smallest for WCBs in the block interaction type."

- 9: caption inconsistent with labels in plot. please correct
  Thank you. The inconsistency between captions and labels has been corrected.

- Pg21, first paragraph: Did you test the sensitivity to your choices?
  We appreciate the reviewer's question. The variable thresholds were chosen to ensure that the composite reflects a dominant interaction type without overly limiting the number of included time steps. We aimed for a balance between signal clarity and sample size, typically selecting at least 5% and at most 25% of winter timesteps for each category. Since the occurrence frequencies of interaction types differ considerably, applying a uniform threshold across all types would be inappropriate. Including significantly more time steps diluted the signal, while more restrictive thresholds reduced interpretability due to insufficient sample sizes. Therefore, a degree of subjectivity was necessary in selecting thresholds to account for these differences.
  We performed sensitivity tests in the North Pacific by varying the threshold and found that while the number of timesteps varied the composite patterns remained qualitatively similar (Fig. R2). This indicates that the main conclusions are not strongly sensitive to the specific thresholds applied. We will add a brief description of this sensitivity analysis to the revised manuscript. We hope this clarification alleviates concerns about the representativeness of our composite methodology.

[Figure]

**Fig. R2:** Sensitivity analysis of thresholds for the interaction types composites. Colors show PV at 320 K at the point of interaction for **(a–c)** ridge, **(d–f)** block, **(g–i)** no interaction, and **(j–l)** cutoff interactions. Columns show (a,d,g,j) lower thresholds (−10% for ridge/no-interaction, −5% for block/cutoff), (b,e,h,k) baseline thresholds as used in the paper, and (c,f,i,l) higher thresholds (+10% for ridge/no-interaction, +5% for block/cutoff). White-green contours mark the 98th and 99th percentiles of WCB air parcel occurrence. The number of time steps included in each composite and the corresponding percentage w.r.t. all DJF time steps in the considered 43-y period are given in each panel.

- L493: I do not follow this speculation. Why would the negative anomaly not simply be the evolving ridge? There is a larger scale positive(!) PV anomaly to the North that may indicate an enhanced large-scale PV gradient, irrespective of WCB activity.

  Thank you for the comment. We agree that the negative PV anomaly may indeed be part of the evolving ridge itself. However, our point is that the WCB outflow likely contributes to this evolution. Since the end-of-ascent of the WCB trajectories is spatially located within the ridge, and we specifically selected timesteps classified as WCB–ridge interactions, we infer that the WCB outflow may enhance the ridge structure and amplify the local PV gradient. While we do not claim the ridge

is solely caused by the WCB, its contribution to the negative PV anomaly and the strengthening of the PV gradient is physically plausible.

- 5.2: The negative anomaly in Fig. 10 evolves only very little during WCB activity. Why do you refer to it the anomaly as a ridge in the beginning of the sequence and a block at the end. To me, this feature very much looks like a block that pre-exists before WCB activity starts.
  Thank you for this observation. We agree that in some cases, the negative PV anomaly shown in Fig. 10 may already exhibit characteristics of a block prior to WCB activity. However, since the interaction type is identified only at the point of interaction, we can only be certain that a block is present at that time, not whether it existed beforehand. Therefore, our composite includes timesteps classified as blocking interactions that may represent a mix of scenarios: in some cases, the block is already established before the WCB outflow, while in others, the WCB outflow may contribute to the development of the blocking structure from a pre-existing ridge.

- 6 is rather long. I suggest introducing subsections "Final discussion" and "Conclusions".
  We appreciate the reviewer's suggestion and will introduce subsections: *Final Discussion* and *Conclusions* to improve readability and flow. Thank you for the helpful recommendation.

- L648: The purpose of this paragraph is not clear to me. Please clarify.
  Thank you for pointing this out. The purpose of this paragraph is to emphasize that both the evolution of the upper-level flow (largely governed by dry dynamics) and the diabatic processes associated with WCB outflows play complementary roles in shaping the flow features. While Sect. 5 emphasizes the importance of the upper-level flow in shaping the different interaction types, this does not diminish the role of WCB outflows. In fact, WCB outflows contribute significantly to the PV anomalies that define these features, even though the features themselves are ultimately modified by the evolving flow. We will clarify this point in the revised manuscript for better understanding.

- L670: I believe that it is worth mentioning at some point that there is much analogy to TC interaction (at this point, e.g., Keller et al. 2019). There, the high sensitivity of the downstream flow can be understood in terms of flow bifurcation points, i.e., without reference to the uncertainty of model microphysics (which in fact is comparatively small).
  We thank the reviewer for this suggestion. Conceptually, we see similarities between the downstream sensitivity of TC interactions and WCB interactions (also because TC interactions often come in the form of WCBs), but more analysis would be required to assess the role of TCs in our climatological study, which is beyond the scope of our paper.

**Reviewer 2**
Vishnupriya et al. present a diagnostic study that classifies warm conveyor belt outflow interactions with the upper-level jet stream (Rossby waveguide). Using ERA5 and Lagrangian tracking, the authors systematically combine multi-decade seasonal climatology of WCB-waveguide interaction types, extending prior case-specific research by introducing an objective classification. The large sample lends confidence that the reported frequencies and patterns are robust climate statistics rather than anecdotal findings. The methodology, from using the well-known LAGRANTO tool for WCB tracking, to identification of waveguide disturbances, is rigorous and consistent with previous literature. The case studies (Fig. 4) and schematic drawing (Fig. 11) are helpful in argumentative demonstrations. Despite the complexity of the subject, the manuscript is generally well-organized and written. Overall, the paper's structure (methods, case examples, climatology, composites, summary) makes it easy to follow the logical progression from methodology to key conclusions. Therefore it has the potential to be published in WCD, but some clarifications and revisions are still needed.

We thank Reviewer 2 for their positive assessment of our study and for recognizing the rigor and clarity of our methodology and presentation. We are glad that the organization and structure of the manuscript facilitate understanding of our approach and findings. We appreciate the reviewer's constructive feedback and will address the suggested clarifications and revisions to further improve the manuscript.

General comments:
● **Threshold Sensitivity.** The classification relies on specific PV threshold criteria that, while grounded in prior studies, are somewhat subjective. For example, ridges are defined by a PV anomaly < −1 PVU and cutoffs by PV < 2 PVU. Likewise, the blocking definition requires a −1.3 PVU anomaly persisting 5 days. It is not fully explored how sensitive the results (especially the relative frequencies of interaction types) are to these threshold values. The authors note that certain methodological elements are subjective but claim the main results are not sensitive to those choices; however, the paper would benefit from evidence of this (e.g. a brief sensitivity test varying the PV anomaly cutoff by some amount). As it stands, it is hard to know if slightly different thresholds might change an event from "weak interaction" to "ridge" or alter the 58.7% ridge frequency.

We thank the reviewer for this important comment regarding the subjectivity of PV-based thresholds. The thresholds we use are based on established approaches documented in prior literature. Specifically, the ridge identification criterion was inspired by Gray et al. (2014), while the blocking threshold (a −1.3 PVU anomaly persisting for at least 5 days) follows well-established definitions used in previous studies (e.g., Schwierz et al., 2004; Croci-Maspoli et al., 2007). Similarly, the tropospheric cutoff definition, isolated PV regions below 2 PVU, is consistent with Wernli and Sprenger (2007) and Sprenger et al. (2017). While we fully agree with the reviewer that the definition of these PV-features is, to a certain degree, subjective, we don't think that adding a sensitivity analysis would increase the quality of our study. In contrast, it might add another level of complexity, which is, however, not essential for the main storyline and the novel aspects of the paper. Since we use well-established approaches (and thresholds), we build on the previous literature and therefore find it acceptable that we avoid adding a sensitivity analysis. We will, however, add a remark that the detailed percentages (e.g., 58.7% ridge frequency) are valid for our specific choice of feature definition, and we mention in the discussion the general caveat of feature-based approaches that they often rely on subjective thresholds.

● **What about the stratosphere?** In the conclusion part (line 585-586), the authors find that in boreal winter, "tropospheric WCB outflows most frequently result in ridge interactions (58.7%), followed by no-interaction (27.7%), block (9.7%), and cutoff (3.9%)interactions." However, such numbers come from a normalization stated in Section 3 (line 250-252): "ridge interacting type (54.0%), followed by no interaction (25.5%), while block (8.9%) and cutoff (3.6%)". So there are about 100% - (54 + 25.5 + 8.9 + 3.6)% = 8% of stratospheric interactions resulting from WCB outflows that are excluded from this study. This 8% is comparable to the 8.9% blocking type and way larger than the 3.6% cutoff type. While it's understandable to focus on tropospheric impacts, these excluded cases (gray dots in their Fig. 2b) are interesting for completeness, especially given that some literatures argue stratosphere-troposphere coupling may be important and modifying blocking frequency (e.g., Davini et al., 2014), which potentially indicates the left-out stratospheric outflows to be a non-negligible category. Hence, it'd be best for the authors to answer if stratospheric WCB outflows are truly negligible in number and in analyses? Also it'd be best for the authors to include the "tropospheric" constraint in line 15 for WCB outflows.

We thank the reviewer for raising this point. In our study, stratospheric WCB outflows were *explicitly excluded* from the classification of interaction types, not because they are not interesting (in fact, they are very interesting when studying troposphere-to-stratosphere transport), but because the mechanisms governing their interaction with the larger-scale flow are distinct and would require a different diagnostic and theoretical approach. Our focus in this manuscript is limited to **tropospheric WCB outflows**, for which the interaction with the midlatitude waveguide can be more directly interpreted in the context of negative PV anomalies related to the WCB outflows, which interact with the PV gradients of the waveguide. We have mentioned this point in the manuscript by noting in Sect. 2.3 that stratospheric outflows are excluded from the interaction classification (line 210).

We agree with the reviewer that this clarification should also be reflected in the Abstract. Accordingly, we will revise the abstract to specify "tropospheric WCB outflows" to avoid ambiguity. In addition, we will also explicitly state in Sect. 2.3 that stratospheric outflows are excluded from the analysis as they fall outside the scope of the present study.

[Figure]

**Fig. R3: (a)** Climatological frequency of occurrence of stratospheric WCB trajectories at their point of interaction (colors, in %) during DJF 1980–2022. Black contours indicate the climatological mean position of the waveguide (2 PVU). **(b)** Corresponding climatological mean outflow age at the point

of interaction (colors, in hours). Black contours indicate the 80th (solid) and 95th (dotted) percentiles of WCB interaction frequency from panel (a).

In response to the reviewer's suggestion, we show the climatological frequency and outflow age of stratospheric WCB outflows (Fig. R3). These events are typically significantly older (in terms of time since the end of ascent), suggesting that these WCB air parcels gained PV gradually, most likely through radiative processes or turbulence.

● **A block could be double counted as a ridge!** Computationally, the authors still enforce mutual exclusiveness by applying a strict hierarchy when they tag each trajectory, so every WCB parcel ends up in one and only one class. Yet, a single trajectory's starting point can coincide with multiple feature types, say a block would typically also satisfy a ridge criteria. This hierarchy, while physically reasoned (a cutoff indicates a more intense wave breaking than a ridge), could lead to ambiguous cases being classified as the highest-ranking feature present. It would be useful to clarify whether the hierarchy ever overrides what a meteorologist might consider the primary interaction. The authors should ensure this automated decision-making doesn't misclassify borderline situations or transitions of types, such as an initial ridge may become a block *after* temporal persistence threshold being met. What's the proportion of such borderline transitions in the study? Wouldn't this also lead to a developed block being slightly older than spawned ridges, in contrast to Fig. 7 panel c, where block is argued to be of the youngest WCB outflow age at the point of interaction? For air parcel in a block, how many of them are of "young age" captured in Fig. 7c, how many of them have stayed in the upper level for more than 5 days (not captured in Fig. 7c)?

We thank the reviewer for this thoughtful comment. Indeed, synoptic features such as ridges, blocks, and cutoffs can spatially and temporally overlap, and a single WCB trajectory can satisfy multiple interaction criteria at a given time. The mutual exclusivity in our classification allows us to distinguish, for example, a WCB interacting purely with a ridge from one embedded in a ridge that has evolved into a block. Without such a hierarchy and mutual exclusiveness, differentiating the characteristics of these interaction types would be challenging.

We agree that these synoptic features can evolve and transition from one interaction type to another, e.g., from ridge to block to cutoff. However, our methodology captures this evolution by classifying interactions independently at each 6-hour time step. For instance, if a WCB trajectory is classified at time t into the ridge category, and this ridge evolves in the next, e.g., 24 h into a block, then the backward trajectories from the block will identify the same WCB trajectory (with an older outflow age), and therefore our approach is capable of handling potentially complex (co)evolutions of PV features and WCBs. The temporal progression of interactions is also reflected in the *outflow age* (Sect. 4.1): younger WCB outflows predominantly interact with ridges and blocks, while older outflows are more frequently associated with cutoffs or no interaction. This supports the interpretation of an evolving interaction pathway over time and suggests a typical evolution pathway from ridge/block interactions toward cutoff or non-interacting states as the outflow ages. In response to the reviewer's suggestion, we show in Fig. R4 the contribution in percentage of young and old outflows for different interaction types. This also confirms that ridge and block interactions are predominantly associated with younger WCB outflows, about 62% and 55% of these interactions, respectively. In contrast, no-interaction and cutoff interactions are dominated by older outflows, with 65–80% of the contributing parcels having outflow ages exceeding 24 h.

[Figure]

**Fig. R4:** The four different WCB–waveguide interaction types: **(a)** no interaction, (**b**) ridge, **(c)** block, and **(d)** cutoff interactions. Contributions from fresh (outflow age less than 24 hours) and old outflows are shown in lighter and darker colors, respectively. The total number of WCB trajectories contributing to each interaction type is indicated in the top-right corner of each panel (M = million).

● **Timing of Interaction Assessment.** The classification is determined at the backward trajectory starting point, which is termed the "point of interaction," but this choice may not capture delayed or downstream impacts. If a WCB outflow does not immediately coincide with a ridge/block at the initial time but contributes to one a day later, such an effect would be missed by the algorithm. In other words, some WCBs might be labeled "no interaction" at the start point but go on to amplify a wave downstream. The paper would benefit from a discussion of this limitation – essentially, the method diagnoses interactions at a fixed time (when trajectories hit the 2-PVU surface) and may not track the subsequent evolution. This is partly addressed by analyzing the eastward advection of no-interaction vs. stagnation of block-type air parcels, but it remains possible that a WCB initially categorized as non-interacting could induce a ridge slightly later. Clarifying how the results might change if the interaction were evaluated over a time window(rather than an instantaneous point) would be useful.

Thank you for this thoughtful comment. We agree that understanding the timing and evolution of WCB–waveguide interactions is crucial. However, we would like to clarify that our methodology does capture the delayed interaction. The classification is applied at every 6-hour timestep in the period considered, allowing us to capture multiple interactions that a single WCB air parcel may experience over time (see also reply to the previous comment). This means that even if a WCB parcel immediately interacts with a ridge or block upon reaching the upper troposphere, any subsequent interaction, e.g., contributing to a cutoff 24 h later, will also be captured in our framework. This temporal evolution is further captured through the concept of *outflow age*, which quantifies the time elapsed since the end-of-ascent.

● **Causality vs Colocation:** The study assumes that if a WCB outflow is co-located with a ridge, block, or cutoff, then the WCB ***interacts*** with the waveguide to produce that feature. This is a reasonable interpretation, but it is essentially inferred rather than directly proven. The authors offer a posteriori justification by noting the much weaker PV anomalies for trajectories classified as non-interacting, implying those indeed had minimal effect. Still, the methodology identifies *associations* between WCB outflows and PV disturbances – it does not demonstrate that the WCB caused the disturbance. It would strengthen the paper if the authors could argue more explicitly that the identified ridges/blocks were actually enhanced by the WCB. As an example, one could ask: might a ridge have existed anyway, with or without the WCB, and the WCB simply happened under it? The implicit assumption of causality could be more critically discussed. This point is important for interpreting the climatology: the paper shows where WCBs *co-occur* with certain waveguide

disruptions, but future work (perhaps using modeling experiments) would be needed to confirm the extent of the WCB's causal influence.

In our terminology, "interaction" refers to the mutual influence between WCB outflow (negative PV anomaly) and the ambient waveguide (PV gradient), acknowledging that the waveguide also acts upon and advects the WCB outflow. We will clarify in the manuscript that the term "interaction" as used here does not imply strict causality. Rather, it denotes the co-occurrence and dynamical linkage between WCB outflow and large-scale waveguide features, without attributing direct cause-effect relationships. It is true that in many cases of, e.g., WCB-ridge interactions, a ridge existed prior to the occurrence of the WCB outflow. Or, in other words, in such situations, there would be also a ridge without the WCB and therefore it would be wrong to imply that the WCB *caused* the ridge. This is not what we imply by using the term WCB-ridge interaction. However, we also have clear evidence from previous studies that ridges are often strongly amplified by WCB outflows (see, e.g., Sect. 5.5 in Wernli and Gray, 2024). A study that shows this effect particularly clearly is the comparison of the idealized dry and moist baroclinic wave simulations by Schemm et al. (2013). Two panels from their Fig. 4 are reproduced here as Fig. R5. The two panels show isentropic upper-level PV in the (left) dry and (right) moist simulation. The moist simulation shows a larger ridge (with lower PV values) in the region of the WCB outflow.

[Figure]

**Fig. R5:** PV on 316 K in idealized baroclinic wave simulations, left for a dry atmosphere, and right for a simulation including moisture. Figure reproduced from Schemm et al. (2013).

For the specific comment about distinguishing pre-existing ridges vs. ridge amplification, we refer to our reply to a similar question from Reviewer 1. We acknowledge that with our approach, we cannot distinguish these two cases, but anyway, in our concept, both situations can be classified as WCB-ridge interaction.

● **Statistical / significance testing.** While the dataset is large, the manuscript does not report formal significance testing for differences between categories. Phrases like "differ significantly" (line 616) in the composite analysis appear to be used qualitatively. It would improve the rigor if the authors could demonstrate that key distinctions (e.g. the differences in outflow latitude or PV between categories) are statistically significant. Similarly, for the composite maps (Fig. 9 and 10), showing stippling or some indication of significance for anomalies would help support statements that, say, blocks are preceded by significantly higher eddy kinetic energy than no-interaction cases. If significance testing was not conducted, the authors should temper the language or clarify that "significantly" is meant in a qualitative sense. Adding some basic statistical analysis would bolster the conclusions that the observed differences are robust and not artifacts of variability.

We thank the reviewer for this suggestion regarding the inclusion of formal significance testing to support our composite analyses. To address this, we will perform Monte Carlo significance tests to quantify the robustness of the key differences between categories, such as anomalies in PV and eddy kinetic energy. Significant regions will be indicated with stippling on composite maps (Figs. 9 and 10), and language referring to "significant differences" will be updated accordingly. We believe this addition will strengthen the robustness of our conclusions.

● **What happens from Mid Ascent to End of Ascent?** One thing stands out in Fig. 6 is that the four types of interactions could hardly be distinguished from each other in panel 6c Mid Ascent, but there are significant spatial differences demonstrated in panel 6b End of Ascent. What in this ascending process is causing the difference? Could you connect Fig. 9 and Fig. 10 arguments with Fig. 6 panel b and c differences? If possible, could you expand in details about the governing mechanism?

We appreciate the reviewer's thoughtful observation, and we too found the contrast between mid-ascent and end-of-ascent in Fig. 6 intriguing. While we do not yet have a definitive explanation and consider a detailed investigation beyond the scope of this study, we speculate that the observed difference may stem from the influence of upper-level dynamics, which likely exert greater control during the late phase of the WCB ascent (mid-ascent to end-of-ascent) compared to the earlier phase (start-of-ascent to mid-ascent).

● **Composite Selection Bias:** The method for constructing composites introduces additional subjective criteria: the authors only composite time steps when a given interaction type is sufficiently dominant (>40% of WCB trajectories in the region). This ensures "pure" cases but might bias the composites towards extreme examples. For instance, a time step with 39% no-interaction, 46% ridge, 10% block, 5% cutoff might be excluded entirely, whereas a time step with 41% no-interaction triggers inclusion as a "no-interaction case". Such hard thresholds (40% for one type, and a secondary 10% cutoff criterion for the cutoff type) could skew the sample of events used for composites. The authors should justify the choice of 40% – presumably to get a decent sample size while maintaining category signal – and perhaps test that varying this threshold (30% vs 50%) does not qualitatively change the composite patterns. If a more inclusive compositing approach yields similar patterns, that would alleviate concern that the composite results are dependent on this filtering.

We appreciate the reviewer's question. The variable thresholds were chosen to ensure that the composite reflects a dominant interaction type without overly limiting the number of included time steps (see our response to a similar question from Reviewer 1).
We performed sensitivity tests in the North Pacific by varying the threshold and found that while the number of timesteps varied, the composite patterns remained qualitatively similar (Fig. R2). This indicates that the main conclusions are not strongly sensitive to the specific threshold applied. We will add a brief description of this sensitivity analysis to the revised manuscript. We hope this clarification alleviates concerns about the representativeness of our composite methodology.

Specific comments:
● **Fig. 4.** The abscissa and ordinate in Fig. 4 are wrong. You cannot have two 80°N in one map, nor 0-60°W being perpendicular to 0-80°E.
The apparent duplication and orientation of coordinates result from the stereographic projection used, which is necessary for visualizing the high-latitude region involved in the case studies.

● **Fig. 8.** Panel a does not explain the letter M in the total number of trajectories.
We appreciate the reviewer's observation. The label "M" refers to "Million," and this has now been explicitly clarified in the figure caption.

● **Fig. 9-10.** The caption of labelling (a,e) as point of interaction is not consistent with the figure labelling (a,e) as start of ascent. Same inconsistency happens for all rows in the plots.
Thank you for noting this inconsistency. We have corrected the figure labels and captions to ensure consistency between the description and the panels.

● **Composites of cutoff-interaction time steps should be included in the manuscript, not in supplementary materials**. The difference for block interactions to ridge of having "intense negative PV anomaly and strong ridge" (line 549) are important, but the cutoff interactions also share this difference and signify a more important sign change of EKE anomaly throughout the ascending time range. I would suggest swap Fig. S15 with Fig.10.
We appreciate the reviewer's suggestion. We agree that the cutoff-interaction composites highlight distinct and relevant features. In response, we will move the cutoff composite figure from the supplementary materials to the main manuscript, as an additional figure.

**References**

Croci-Maspoli, M., Schwierz, C., and Davies, H. C.: A multifaceted climatology of atmospheric blocking and its recent linear trend, J.Climate, 20, 633–649, https://doi.org/10.1175/JCLI4029.1, 2007.

Grams, C. M., Magnusson, L., and Madonna, E.: An atmospheric dynamics perspective on the amplification and propagation of forecast error in numerical weather prediction models: A case study, Q. J. Roy. Meteor. Soc., 144, 2577–2591, https://doi.org/10.1002/qj.3353, 2018.

Gray, S. L., Dunning, C. M., Methven, J., Masato, G., and Chagnon, J. M.: Systematic model forecast error in Rossby wave structure, Geophys. Res. Lett., 41, 2979–2987, https://doi.org/10.1002/2014GL059282, 2014.

Joos, H. and Forbes, R. M.: Impact of different IFS microphysics on a warm conveyor belt and the downstream flow evolution, Q. J. Roy. Meteor. Soc., 142, 2727–2739, https://doi.org/10.1002/qj.2863, 2016.

Madonna, E., Wernli, H., Joos, H., and Martius, O.: Warm conveyor belts in the ERA-interim dataset (1979–2010). Part I: climatology and potential vorticity evolution, J. Climate, 27, 3–26, https://doi.org/10.1175/jcli-d-12-00720.1, 2014.

Madonna, E., Boettcher, M., Grams, C. M., Joos, H., Martius, O., and Wernli, H.: Verification of North Atlantic warm conveyor belt outflows in ECMWF forecasts, Q. J. Roy. Meteor. Soc., 141, 1333–1344, https://doi.org/10.1002/qj.2442, 2015.

Schemm, S., Wernli, H., and Papritz, L.: Warm conveyor belts in idealized moist baroclinic wave simulations, J. Atmos. Sci., 70, 627–652, https://doi.org/10.1175/JAS-D-12-0147.1, 2013.

Schwierz, C., Croci-Maspoli, M., & Davies, H. C.: Perspicacious indicators of atmospheric blocking. Geophysical Research Letters, 31, 6. https://doi.org/10.1029/2003gl019341, 2004.

Sprenger, M., Fragkoulidis, G., Binder, H., Croci-Maspoli, M., Graf, P., Grams, C. M., Knippertz, P., Madonna, E., Schemm, S., Škerlak, B., and Wernli, H.: Global climatologies of Eulerian and Lagrangian flow features based on ERA-Interim, Bull. Amer. Meteor. Soc., 98, 1739–1748, https://doi.org/10.1175/BAMS-D-15-00299.1, 2017.

Wernli, H. and Gray, S. L.: The importance of diabatic processes for the dynamics of synoptic-scale extratropical weather systems – a review, Weather Clim. Dynam., 5, 1299–1408, https://doi.org/10.5194/wcd-5-1299-2024, 2024.

Wernli, H. and Sprenger, M.: Identification and ERA-15 climatology of potential vorticity streamers and cutoffs near the extratropical tropopause, J. Atmos. Sci., 64, 1569–1586, https://doi.org/10.1175/JAS3912.1, 2007.

---

## Author Response (AR2)

**Author's response for egusphere-2025-1731**

**The interaction of warm conveyor belt outflows with the upperlevel waveguide: a four-type climatological classification**

by Selvakumar Vishnupriya, Michael Sprenger, Hanna Joos, and Heini Wernli Aug 29, 2025

The following section presents the reviewers' comments in **blue** and our responses in **black**.

**Reviewer 1**

First of all, I thank the authors for their appreciative style in replying to my comments. Furthermore, I appreciate that the authors have responded in a constructive and thoughtful manner, which clarifies the authors' perspectives (for me, as a reviewer, but more importantly also in the manuscript for future readers).

To me personally, it is still conceptually more appealing to think of the overall problem in terms of a disturbance of the "waveguide" by the WCB's divergent outflow and then, in a second step, of how this disturbance subsequently evolves, rather than combing the "divergent forcing" and the subsequent (nonlinear) evolution of associated PV anomalies into the metric under consideration. Of course, the authors are free to take their "combined" perspective. The authors have clarified their perspective in the revised version, and I am happy to recommend the manuscript for publication after consideration of the few minor comments below. Kind regards

We sincerely thank the reviewer for the positive and encouraging feedback on our revised manuscript and response letter. We truly appreciate your engagement and invaluable suggestions that improved our work.

We acknowledge the reviewer's point that a two-step framing, divergent forcing followed by nonlinear PV evolution, may provide a more intuitive representation of the problem. While in our study, we deliberately adopted a combined perspective to capture both the direct forcing and the subsequent evolution within a unified framework, we agree that the reviewer's framing is equally valid and insightful. We are grateful that the revised manuscript now meets the reviewer's expectations and appreciate the recommendation for publication. Below, we address the remaining minor comments in detail.

**Minor comments:**

"interaction intensity & interaction types": In the revised version, these terms are now more explicitly defined and better motivated. I have no issues with the authors' definition. To me, however, the terms "interaction intensity" and "interaction types" reflect their definitions poorly at best; in the worst case the terminology is misleading and confusing to future readers. I thus recommend that the authors re-consider this terminology to further improve their manuscript.

We thank the reviewer for this thoughtful comment. We acknowledge that terminology in this context can be challenging, as any choice may have limitations in fully capturing the underlying concepts and being clear and intuitive for the future readers. However, after careful consideration (we discussed terminology a lot during our study!), we prefer to retain the terms "interaction intensity" and "interaction types", as they provide a consistent framework throughout the manuscript. We have taken care to define and explain these terms in detail where they first appear in the paper to avoid ambiguity, and we believe that the additional explanations we added in the revised version will enable readers to understand the intended meaning.

In response to my previous comment on the concept of age of outflow, the authors have revised their manuscript (L369-L372). In this revision, the language used implies causality, i.e., how the WBC influence the larger-scale flow. Please revise this revision according to your reply and revisions made in response to my first general comment of the previous review.

Thank you for pointing this out. We have revised the sentence at Line 371 from "...provides insight into the evolving influence of WCB outflows on the large-scale flow" to "...provides insight into the evolution of WCB outflows within the large-scale flow", to avoid implying causality and to ensure consistency with our clarified framing of interaction.

This last comment is rather some minor food for thought for the authors and does not imply an action item. The comment refers to the authors' response to my second last (specific) comment of the previous review (which referred to what is now the first paragraph of 6.2). There, the authors write that "the upper-level flow" is "largely governed by dry dynamics". A statement that seems to contradict the authors' response to my specific comment on their L32, and the review paper by Wernli and Gray. Is the upper-level, midlatitude flow largely governed by dry dynamics or do moist process play an important role? It seems to me that both statements cannot be true at the same time. (The authors' statement in the response is inconsequential for the manuscript and I am happy with the authors' modification made to the manuscript.)

We thank the reviewer for raising this important point. We agree that our phrasing in the response could have been misleading. What we intended to express is that most aspects of the upper-level midlatitude flow can be interpreted and understood to first order within the framework of dry dynamics, while moist processes add an important additional layer of complexity. This does not imply that moist dynamics are unimportant; on the contrary, as highlighted by Wernli and Gray (2025), they play a crucial role in shaping and amplifying the flow.

**Reviewer 2**

Vishnupriya et al. have successfully addressed the comments I've given out, and have implemented major revisions including statistical testing and sensitivity analysis for robustness of their warm conveyor belts (WCB) outflow analyses. Careful wording has been done to avoid causality suggesting interaction, and to rather highlight the two-way influence / co-occurence between the upper-level troposphere and WCB outflows. Here are some minor changes needed to make the paper suitable for publication.

We sincerely thank the reviewer for the positive evaluation of our work and for acknowledging the major revisions we have implemented. We are grateful for the constructive feedback throughout the review process, which has helped us to substantially improve the manuscript. We are pleased that the reviewer considers the manuscript suitable for publication. Below, we address the remaining minor comments in detail.

**Minor comments:**

**WCB outflow classification.**

The authors have clearly clarified the possibility of each WCB outflow trajectory transitions, based on their 6-hourly updates of classifications, and the Figure R4 is very helpful in reading out the composition of WCB parcels in blocking by "age".

With this mutual exclusiveness with progression in time, what you are really referring to is that "blocking but whose part hasn't been anomalous enough to be a cut-off (high)" would have younger WCB parcels making up to 55.6%. The parcel could have still stayed within the block for longer, but just transitioned to be categorized as a cutoff by this progressive classification.

If we think from a meteorological common sense - a true blocking, taking your Figure 2b's Ural blocking as an example, would be separated into (1) block (minus cutoff) and (2) cutoff parts by your definition; the block (minus cutoff) part would have more young WCB parcels, but the cutoff (the majority area of this Ural blocking!) would have way less young WCB parcels. This actually answers my confusion of "[why am I not seeing] a developed block being slightly older than spawned ridges" - you are just referring to the small edges of the block as block (minus cutoff), and the older parcels might just sit in the cutoff classification in the blocking center. Is there a nicer way to avoid using the misleading "block" that the audience might mistake it for the entire region of the blocking high? We thank the reviewer for the detailed feedback. The transition from block to cutoff interactions occurs only in a subset of cases, as cutoff interactions are much less frequent than block interactions. The differing spatial hotspots of block and cutoff features (Fig. 1b,c) further confirm that these are largely distinct. The point-of-interaction regions of block interactions (Fig. 5c) exhibit similar hotspots to those of block features (Fig. 1b), indicating that the classification effectively captures the majority of block interactions.

The reviewer's concerns are already addressed in the manuscript: (i) we explain that interacting WCB parcels do not cover the entire weather feature, ridge, block, or cutoff since we focus only on the WCB parcels within the features (L286–290, 300–303), (ii) we elaborate on the classification hierarchy, which enforces mutual exclusivity, in Sections 2.3 and 2.4. We believe these clarifications will help future readers accurately interpret our classification methodology.

**Preexisting ridge vs new ridge - co-occurence and causality**

The authors have clarified about their interaction to be co-occurrence of WCB outflows with the classified features, and have edited the manuscript to incorporate the situation of WCB outflows

injecting into preexisting ridges by L392-393: "the WCB trajectories contribute to the formation or maintainance". There's a spelling error: the noun form of "maintain" is "maintenance", not "maintainance".

We thank the reviewer for pointing out the typo. We have corrected "maintainance" to "maintenance" in the revised manuscript (L394).

**Stratosphere WCB outflows.**

The authors have correctly edited the classification ratio in the abstract to be conditioned on troposphere only, and have included Figure R3 as a preliminary examination of the stratospheric WCB outflow. Figure R3 could thus be placed in the supplementary materials and refer to it in manuscript L218, or in section 6.2 as a possible future study direction.

We thank the reviewer for the suggestion. We agree that the preliminary examination of stratospheric WCB outflows is interesting, and we have included Fig. R3 in the supplementary materials (Fig. S4) with reference to it in the manuscript (L218).

**Figure 11 caption: "np-interaction" should be "no-interaction".**

We thank the reviewer for carefully checking our figure captions. The caption of Figure 11 has been corrected from "np-interaction" to "no-interaction."